# PromptWise: Extending Contextual Bandits to Multi-Round Prompt Routing for Cost-Efficient LLM Inference

## Abstract

The rapid advancement of generative AI has provided users with a wide range of well-trained models to address diverse prompts, with substantial variation in both performance and service cost. When selecting a model for a given prompt, users should therefore consider not only expected output quality but also the monetary cost of querying the model. Many existing approaches assume fixed or single-shot model assignment per prompt, whereas practical deployments may require online adaptation and cost accumulation across multiple attempts. In this paper, we introduce PromptWise, an online learning framework for cost-aware prompt assignment that learns from sequential interactions between prompts and models. Prompt-Wise maintains and continuously updates estimates of prompt–model compatibility based on observed feedback, and uses these estimates to make adaptive, cost-sensitive decisions under uncertainty. Unlike standard contextual bandits that make a one-shot decision per prompt, PromptWise employs a cost-aware contextual bandit structure that allows sequential model assignments per prompt and explicitly accounts for the cumulative service cost incurred across multiple attempts. Through numerical experiments on tasks such as code generation and translation, we demonstrate that PromptWise achieves performance comparable to baseline selection methods while substantially reducing overall service cost.

## 1 Introduction

Recent advances in large language models (LLMs) and text-guided generative AI have expanded access to a broad array of services (Rombach et al., 2022; Touvron et al., 2023; OpenAI et al., 2024; DeepSeek-AI et al., 2025; Gemini-Team et al., 2025). Users can now select among many well-trained models to address diverse prompts. This abundance raises a fundamental question: How should one assign prompts to models in a principled and accurate manner?

A straightforward strategy is to choose the model with the highest averaged performance score and apply it universally to all prompts. However, the recent literature highlights that performance may vary significantly across prompt categories. A model may excel on one class of prompts yet perform suboptimally on others (Qin et al., 2024; Frick et al., 2025; Hu et al., 2025). Such discrepancies arise from differences in training data and architectural choices. To exploit this heterogeneity, prior studies propose either offline selector models (Qin et al., 2024; Chen et al., 2023; Frick et al., 2025) or online bandit algorithms (Hu et al., 2025) that adaptively allocate prompts to specialized models.

While these learning-based approaches improve prompt-specific performance, they all assume each prompt is *assigned to exactly one model*, overlooking the role of monetary and computational costs in users' decisions on how to assign prompts to available models. As also discussed by Chen et al. (2023), in practice, users are not constrained to single-shot decisions. They may query multiple models for the same prompt, escalating to more expensive options only if cheaper attempts fail, as long as the cumulative expense is justified.

For instance, consider a user who chooses among OpenAI's `GPT-5.5-pro` ($180 per million tokens (PMT)), `GPT-5.4` ($15 PMT), and `GPT-5.4-mini` ($4.5 PMT)[1] to solve Sudoku puzzles (illustarted in Figure 1). A

---

[1] https://platform.openai.com/docs/pricing

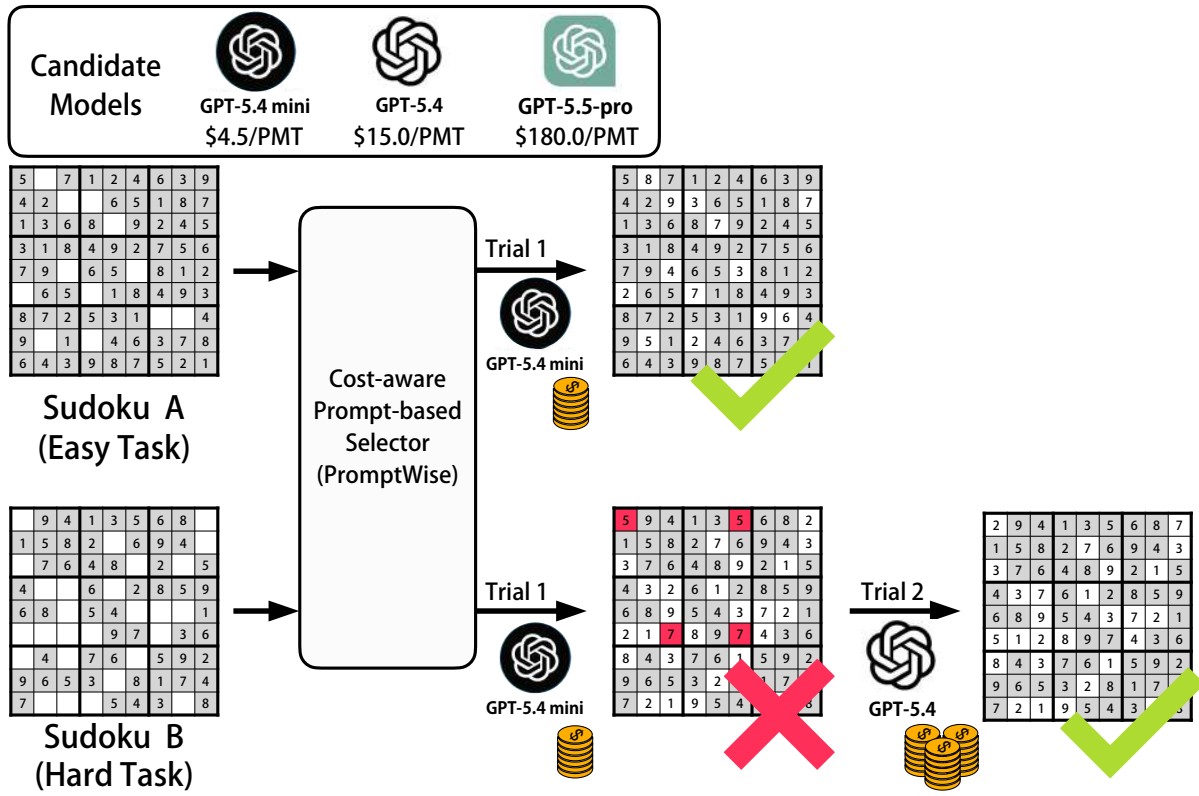

Figure 1: Illustration of cost-aware prompt assignment in Sudoku solving, where a single prompt may be routed to multiple models. PromptWise, via online learning, would route easier puzzles to the inexpensive `GPT-5.4-mini`, while for harder instances it may escalate to stronger and more expensive available models `GPT-5.4` and `GPT-5.5-pro`.

seemingly cost-efficient strategy would be to first attempt the least expensive `GPT-5.4-mini`, escalate to `GPT-5.4` if unsuccessful, and resort to `GPT-5.5-pro` only for the hardest puzzles. Although the cheapest model may fail on complex tasks with high probability, the 100 times price difference could justify probing it first. Such multi-assignment strategies are natural in real-world deployments, yet absent from existing online model-selection frameworks.

In this work, we introduce *PromptWise*, an online learning framework for cost-aware online prompt-to-model assignment. At its core, PromptWise is a cost-aware contextual multi-armed bandit (CA-CMAB) formulation that explicitly incorporates both pricing heterogeneity and the possibility of multiple assignments per prompt, which differentiates PromptWise from the non-online-learning framework of FrugalGPT (Chen et al., 2023) and Prompt-to-Leaderboard (Frick et al., 2025) to address the task. Unlike standard contextual bandits (Langford & Zhang, 2007; Li et al., 2010; Chu et al., 2011), which commit to a single decision per round, CA-CMAB tracks both the best achieved performance per prompt and the cumulative cost of all attempts (illustrated in Figure 2).

Specifically, PromptWise extends the classical Upper Confidence Bound (UCB) framework (Auer, 2000; Bubeck & Cesa-Bianchi, 2012) with a new sum-min regret formulation. For each prompt, only the best outcome across attempted models contributes to the performance regret, while costs are fully accumulated. This structure formalizes the trade-off between predictive reward and monetary expenditure, bridging the gap between theoretical online learning formulations and the realities of commercial generative AI services.

We evaluate PromptWise through both controlled simulations, where ground-truth optimal strategies are known, and real-world generative AI tasks, including Sudoku and chess puzzles as well as code generation and translation benchmarks (HumanEval (Chen et al., 2021), HumanEval-X (Zheng et al., 2023),

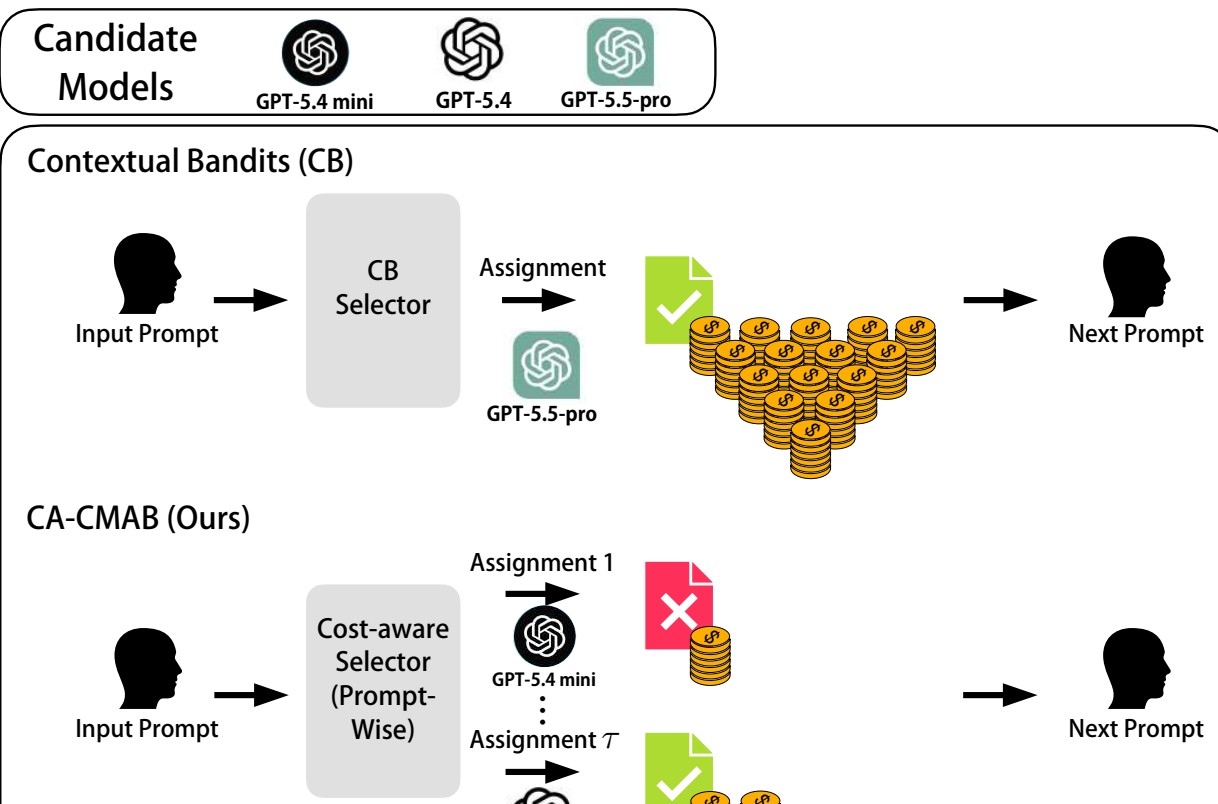

Figure 2: Interaction protocol of standard contextual bandits (top) versus our cost-aware contextual multi-armed bandit (CA-CMAB) framework in PromptWise (bottom). Unlike the standard setting, CA-CMAB enables multiple model assignments per prompt and explicitly balances accuracy (reward) with cumulative service cost, better reflecting practical usage scenarios.

BigCodeBench (Zhuo et al., 2024)). Across settings, PromptWise rapidly adapts to prompt distributions, achieves accuracy comparable to cost-unaware baselines, and reduces expenditures by large margins. In summary, this work's contributions are threefold:

- We formalize the cost-aware multi-assignment prompt-selection problem and highlight its practical importance for generative AI services.

- We propose PromptWise, a CA-CMAB algorithm that balances performance and cost via a novel sum-min regret metric.

- We demonstrate through simulations and real-world benchmarks that PromptWise achieves substantial cost reductions without sacrificing accuracy.

## 2   Related Work

**Generalized linear contextual bandits.** Contextual bandit (CB) is a widely used framework for sequential decision-making with side information, which has been extensively used in recommendation systems and advertising. A relevant assumption in the CB setup is that the expected reward conditioned to an incoming context admits a generalized linear model (GLM) (Filippi et al., 2010; Li et al., 2017). Particularly,

the logistic bandit framework (Faury et al., 2020; Dong et al., 2019) has demonstrated stronger performance in applications where rewards are binary.

**Bandit problems with cost constraints.** A line of study considers multi-armed bandits (MAB) in a cost-aware setup, where each arm is associated with a cost distribution. The learner aims to not only optimize the cumulative return but also minimize the cost of pulling the arms. Ding et al. (2013) study cost-aware bandits with a constraint on the total paying cost. In Multi-fidelity MAB (Kandasamy et al., 2016; Wang et al., 2023), the learner may choose to observe (pseudo-)rewards of an arm at some fidelity level and incur the corresponding cost. Recently, Kanarios et al. (2024) study cost-aware best-arm identification, where the goal is to identify the largest reward arm using the minimum expected cost. To the best of our knowledge, cost-aware bandit algorithms in the literature consider optimizing a linear utility, e.g., the cumulative return minus (a regularized) cumulative cost.

**Online learning methods for training and selecting generative models.** A line of research incorporates online learning and reinforcement learning in the training and selecting generative models. Grnarova et al. (2017) formulate the problem of training a Generative Adversarial Network (GAN) as finding a mixed strategy in a zero-sum game. They propose an online learning-based training algorithm, which provably converges to an equilibrium. Daskalakis et al. (2018) show that optimistic mirror descent (OMD) can address the limit cycling issue in training GANs. More recent work proposes the reinforcement learning from human feedback (RLHF) pipeline for training large language models (LLMs) (Ouyang et al., 2022; Rafailov et al., 2023; Munos et al., 2024; Dong et al., 2024) and image-based generative models (Yuan et al., 2023; Xu et al., 2023). Regarding the selection of generative models, recent work proposes online learning-based algorithms (Hu et al., 2025) to a optimize performance score among a group of generative models. Wang et al. (2025) considers finding a routing policy among a group of LLMs to maximize a linear utility of the response quality and the model cost.

**Offline learning approaches for cost-aware LLM selection.** A complementary line of work studies *offline-trained* or *static* strategies for reducing inference cost by routing each prompt to one or more LLMs using supervision collected in advance. FrugalGPT (Chen et al., 2023) utilizes an *LLM cascade* in which a query is routed through a sequence of models from cheaper to stronger ones, using a generation-quality scorer and a stopping ("judger") mechanism to decide whether to accept the current answer or escalate to a more expensive model. Prompt-to-Leaderboard (P2L) (Frick et al., 2025) learns prompt-conditional Bradley–Terry coefficients from large-scale human preference data. RouteLLM (Ong et al., 2024) similarly learns routing policies from preference data, aiming to balance response quality with cost by dynamically selecting between a stronger, expensive model and a weaker, cheaper model, while emphasizing single-model invocation per query to control cost and latency.

## 3 Cost-Aware Contextual Multi-Armed Bandit (CA-CMAB) Settings

**Interaction protocol.** In this section, we introduce the framework of Cost-Aware Contextual MAB (CA-CMAB). The interaction protocol is given in Protocol 1. At each *step* $t \in [T]$, where $[T] := \{1, 2, \cdots, T\}$, a *context* $x_t \in \mathcal{X} \subseteq \mathbb{B}^d := \{x \in \mathbb{R}^d : \|x\|_2 \leq 1\}$ (e.g., the input text prompt) is drawn from the contextual distribution $p_0$. Different from the standard contextual bandit setting (Langford & Zhang, 2007), the learner can interact under the context $x_t$ for multiple *rounds* by taking a sequence of actions in $\bar{\mathcal{A}} := \{a_0\} \cup \mathcal{A}$, where $a_0$ is the *null action*, i.e., the user decides to terminate the process for prompt $x_t$ and moves to the next prompt, and $\mathcal{A}$ is the set of *arms*.

Specifically, after pulling an arm $a_{t,i} \in \mathcal{A}$ at the $i$-th round, the learner observes a reward $r_{t,i} \sim R_{a_{t,i}}(x_t)$ and pays a fixed cost $c_{t,i} := c_{a_{t,i}}$. The environment moves to the next task if the learner chooses the null action $a_0$ or the allocated round budget is completely used up (i.e., $i = \tau_{\max}$). We denote by $\tau_t \in [\tau_{\max}] := \{0, 1, \cdots, \tau_{\max}\}$ the (random) number of arms pulled within step $t$. For convenience, we define $a_{t,\tau_t+1} =: a_0$. Note that we assume that taking the null action yields a zero reward and cost, i.e., $R_{a_0} = c_{a_0} = 0$, which means that the action of moving to the next prompt costs nothing and also does not provide any reward. In this paper, we exclusively focus on the scenario where the rewards are *binary*; however, the bandit framework can further extend to settings with non-binary rewards.

---

**Protocol 1** Cost-Aware Contextual MAB (CA-CMAB)

---

**Require:** step $T$, context distribution $p_0$, set of actions $\bar{\mathcal{A}} = \{a_0\} \cup \mathcal{A}$, cost $\{c_a\}_{a \in \bar{\mathcal{A}}}$, reward distributions $\{R_a\}_{a \in \bar{\mathcal{A}}}$, cost parameter $\lambda > 0$, round budget $\tau_{\max} \in \mathbb{N}_+$.

1: **for** step $t = 1, 2, \cdots, T$ **do**
2:     The environment draws context $x_t \sim p_0$.
3:     Initialize the counter $\tau_t \leftarrow 0$.
4:     **while** $\tau_t < \tau_{\max}$ **do**
5:         Select action $a_{t, \tau_t+1} \in \mathcal{A} \cup \{a_0\}$.
6:         **If** $a_{t, \tau_t+1} = a_0$ **then break.**
7:         Increment the counter $\tau_t \leftarrow \tau_t + 1$
8:         Incur cost $c_{a_{t, \tau_t}}$.
9:         Observe reward $r_{t, \tau_t} \sim R_{a_{t, \tau_t+1}}(x_t)$.
10:        **If** $r_{t, \tau_t} = 1$ **then break.**
11:     **end while**
12: **end for**

---

**Assumption 1** (Binary reward). For each arm $a \in \mathcal{A}$, the binary reward distribution follows $\mathrm{Ber}(q_a(x))$ given the context outcome $x \in \mathcal{X}$, where the success probability $q_a(x) \in [0, 1]$.

*Remark* 1. The assumption of binary reward has been widely considered in the bandit literature, including Bernoulli bandits (Berry, 1972; Agrawal & Goyal, 2012) and logistic bandits (Dong et al., 2019; Faury et al., 2020). We note that a binary reward suits applications with success/failure outputs, such as puzzle-solving (correct or incorrect solution) and code generation (passing all the test cases or not).

**Policy and cost-aware utility.** The learner interacts with the environment for $T$ steps. Upon receiving context $x_t \in \mathcal{X}$ at step $t$, the learner plays (a sequence of) actions according to some policy $\pi_t$, which may depend on the previous observations. The learner aims to find a policy that maximizes the following *utility* function, which trades off between the maximum obtained reward and the cumulative paid cost, where $\lambda \geq 0$ is called the *cost coefficient* parameter:

$$\pi^\star(x_t) := \arg\max_\pi u^\pi(x_t), \text{ where } u^\pi(x_t) := \mathbb{E}_\pi \left[ \max_{i=1,\cdots,\tau_t} r_{t,i} - \lambda \sum_{i=1}^{\tau_t} c_{t,i} \right] \tag{1}$$

*Remark* 2 (Comparison to contextual bandits). Figure 2 highlights the key difference between our proposed CA-CMAB framework and the standard contextual bandit (CB) setting. When applying a standard CB algorithm to the prompt assignment task, the algorithm is supposed to have only one model selection per prompt. On the other hand, CA-CMAB allows the learner to act under a context multiple times before moving to the next task (i.e., prompt). Therefore, the CA-CMAB framework allows a cost-aware strategy to attempt the less expensive options even if they have a marginal chance of success, since a likely failure outcome can be followed by the selection of more expensive models to address the same prompt. However, the one-round interaction of standard CB for each prompt limits the bandit algorithm to attempt models with higher success chance, since the learner is supposed to move to next prompt in the subsequent round.

Under the assumption of binary rewards and an unlimited budget for each round, we characterize the optimal policy in the following proposition. The proof is in Appendix A.1.

**Proposition 1** (Oracle). *Under Assumption 1 and with unlimited round budget, an optimal policy (1) is as follows: for any incoming context $x \in \mathcal{X}$, it takes action $a^\star(x)$ if a reward of 1 is not observed, which is given by*

$$a^\star(x) = \begin{cases} a_0, & \text{if } \max_{a \in \mathcal{A}}\{q_a(x) - \lambda c_a\} \leq 0, \\ \arg\min_{a \in \mathcal{A}} \frac{c_a}{q_a(x)}, & \text{otherwise,} \end{cases} \tag{2}$$

where $q_a(x)$ and $c_a$ are the success probability conditioned to context $x$ and the (fixed) cost for any arm $a \in \mathcal{A}$, respectively. The expected total cost of this policy is $\frac{c_{a^\star(x)}}{q_{a^\star(x)}}$, and the expected utility is given by

$$u^\star(x) = \mathbb{1}\left[a^\star(x) \neq a_0\right] \cdot \left(1 - \lambda \cdot \frac{c_{a^\star(x)}}{q_{a^\star(x)}(x)}\right). \tag{3}$$

## 4 The PromptWise Algorithm

In this section, we propose PromptWise as an online learning algorithm to address the prompt-based model selection task in the CA-CMAB setting. Given the input prompt playing the role of context variable, PromptWise selects the action according to 2 according to the estimated success probabilities for every arm, i.e., the prompt-based generative model. As the success probabilities are estimated using the limited previous observations and could be different from the true success probability (based on the true underlying distribution of the models), directly plugging them into Equation (2) can lead to a sub-optimal choice of generative models. To address the issue in the online selection scenario, our proposed PromptWise applies the Upper Confidence Bound (UCB) approach (Auer, 2000; Bubeck & Cesa-Bianchi, 2012) widely used in the bandit literature, to avoid sub-optimal model assignment policies in the online selection process.

### 4.1 Overview of PromptWise

We now introduce PromptWise (presented in Algorithm 1). In the exploration phase (Lines 1-4), PromptWise plays every arm for $\tau_{\exp}$ initial rounds, where $\tau_{\exp} \in \mathbb{N}_+$ is a tuning parameter. Upon observing the reward values, PromptWise calculates the maximum-likelihood estimation (MLE) of the weight vectors $\{\theta_a^\star : a \in \mathcal{A}\}$, which is given by

$$\widehat{\theta}_a := \operatorname*{argmax}_{\theta \in \mathbb{R}^d}\left\{\widehat{\ell}_a(\theta) := \sum_{(X,R) \in \mathcal{D}_a}\left[R \cdot \log \mu\big(\langle \theta, X \rangle\big) + (1-R) \cdot \log\big(1 - \mu(\langle \theta, X \rangle)\big)\right]\right\}. \tag{4}$$

The MLE is utilized to estimate the success probabilities (Lines 3 and 6). At any subsequent step (Lines 5-17), PromptWise interacts with the environment by attempting to compute the oracle action (2) on a set of estimated success probabilities (Lines 7-15). We include an exploration bonus when computing the estimate $\widehat{q}_a$, which is given by

$$\widehat{q}_a := \mu\left(x_t^\top \widehat{\theta}_a + \alpha \|x_t\|_{V_a^{-1}}\right), \text{ where } V_a = \sum_{(X,R) \in \mathcal{D}_a} XX^\top. \tag{5}$$

The design of the exploration bonus follows the UCB methods for the generalized linear contextual bandits (Filippi et al., 2010; Li et al., 2017), and it can be shown that $\widehat{q}_a$ upper bounds the groundtruth value $q_a(x_t)$ with a high probability. To interact with the environment, PromptWise attempts to predict the optimal action (2) by replacing the groundtruth success probabilities with their UCB estimates (5). Specifically, PromptWise takes the null action $a_0$ and moves on to the next step if all the arms are predicted to fail to address the task (Line 9). Otherwise, PromptWise plays the arm that yields the highest predicted utility (Line 11). After receiving a new observation, PromptWise performs inner-step updates to both the MLE and the estimated success probability for the played arm (Line 14). Hence, PromptWise can utilize the knowledge of the observations in the earlier rounds to adjust the arm selection in the subsequent rounds.

Next, we establish a theoretical regret bound for EP-PromptWise (Epoch-PromptWise), a simplified variant of the PromptWise algorithm that updates the model selection strategy only once at the beginning of each prompt's assignment process. This restricted update rule yields a more analytically tractable formulation while preserving the core exploration–exploitation structure of PromptWise. Under standard stochastic assumptions, we show that EP-PromptWise achieves sublinear regret. The formal regret guarantee is stated in Theorem 2 in the Appendix. We further present numerical results to compare the two algorithms, which show that PromptWise can adjust model adaptively for the input prompt and achieve better performance (See Appendix C.1).

---

**Algorithm 1** PromptWise

---

**Require:** step $T$, cost parameter $\lambda > 0$, regression dataset $\mathcal{D}_a \leftarrow \varnothing$ for $a \in \mathcal{A}$, $V_a \leftarrow \mathbf{0}_{d \times d}$, round budget $\tau_{\max}$, tuning parameter $\tau_{\exp}$ and $\alpha$

1: **for** each arm $a \in \mathcal{A}$ **do**
2:      Collect samples $\mathcal{D}_a \leftarrow \mathcal{D}_a \cup \{(x_s, r_s)\}_{s=1}^{\tau_{\exp}}$ by pulling arm $a$ (and moving to the next context) upon observing contexts $\{x_s\}_{s=1}^{\tau_{\exp}}$, which are drawn from the environment.
3:      Compute the MLE $\widehat{\theta}_a$ defined in Equation (4) by gradient ascent.
4: **end for**
5: **for** step $t = |\mathcal{A}|\tau_{\exp} + 1, \cdots, T$ **do**
6:      Observe context $x_t$ and compute the estimated success probability $\widehat{q}_a$ for each arm $a \in \mathcal{A}$, which is given by Equation (5).
7:      **while** not playing null action $a_0$ **do**
8:          **if** $\max_{a \in \mathcal{A}}\{\widehat{q}_a - \lambda \cdot c_a\} \leq 0$ **or** hitting the round budget **or** receiving a reward of 1 **then**
9:              Play null action $\widehat{a} \leftarrow a_0$.
10:          **else**
11:              Play arm $\widehat{a} \leftarrow \arg\min_{a \in \mathcal{A}} \frac{c_a}{q_a}$.
12:              The environment draws a reward $r$.
13:              Update dataset $\mathcal{D}_{\widehat{a}} \leftarrow \mathcal{D}_{\widehat{a}} \cup \{(x_t, r)\}$.
14:              Compute the MLE $\widehat{\theta}_{\widehat{a}}$ and estimate $\widehat{q}_{\widehat{a}}$.
15:          **end if**
16:      **end while**
17: **end for**

---

**Theorem 1** (Regret of variant PromptWise). *Let the choice of parameters $\tau_{\exp}, \alpha$ and round budget $\tau_{\max}$ follows from Theorem 2. With probability at least $1 - 2\delta$, the regret of running PromptWise without inner-step updates (Algorithm 2 in the Appendix) for $T$ steps is bounded by $|\mathcal{A}|\tau_{\exp} + \tilde{O}(d\sqrt{|\mathcal{A}|T})$.*

### 4.1.1 Extension to kernel methods

The original PromptWise algorithm utilizes a linear function (inside the logistic function) to predict the success probability. To further improve the predictions, one approach is to incorporate non-linearity using kernel methods. In the following, we propose PromptWise-KLR (see Algorithm 3) as an extension of PromptWise to kernel-based prediction functions. Let $k : \mathbb{R}^d \times \mathbb{R}^d \to \mathbb{R}$ denote a kernel function, e.g., the radial basis function (RBF) kernel $k_{\mathrm{RBF}}^\sigma(x, x') = \exp(-\frac{\|y - y'\|_2^2}{2\sigma^2})$. Given the regression dataset $\mathcal{D}_a$, we denote by $\boldsymbol{K}_{\mathcal{D}_a} := [k(x, x')]_{(x,r),(x',r') \in \mathcal{D}_a} \in \mathbb{R}^{|\mathcal{D}_a| \times |\mathcal{D}_a|}$ the Gram matrix, and the corresponding (squared) matrix norm is given by $\|w\|_{\boldsymbol{K}_{\mathcal{D}_a}}^2 := w^\top \boldsymbol{K}_{\mathcal{D}_a} w$ for every $w \in \mathbb{R}^{|\mathcal{D}_a|}$. Let $\beta \geq 0$ denote a tuning parameter. PromptWise-KLR leverages kernel logistic regression (KLR) (Zhu & Hastie, 2001; Wahba et al., 2018) to predict the success probability for the incoming context $x_t$:

$$\widehat{q}_a(x_t) = \mu\left( \sum_{(X,R) \in \mathcal{D}_a} \widehat{w}_a^X k(X, x_t) + \alpha \cdot \mathcal{B}(x_t | \mathcal{D}_a) \right),$$

where $\mathcal{B}(x_t | \mathcal{D}_a)$ is the bonus term and the weight $\widehat{w}_a := [\widehat{w}_a^X]_{(X,R) \in \mathcal{D}_a}^\top \in \mathbb{R}^{|\mathcal{D}_a|}$ is solved from minimizing the (regularized) negative log-likelihood:

$$\widehat{w}_a := \underset{w \in \mathbb{R}^{|\mathcal{D}_a|}}{\arg\min} \ \ell(w),$$

where we consider the objective function:

$$\ell(w) = -\sum_{(X,R) \in \mathcal{D}_a} \left[ R \cdot \log \hat{p}_a(X, w | \mathcal{D}_a) + (1 - R) \cdot \log(1 - \hat{p}_a(X, w | \mathcal{D}_a)) \right] + \beta \|w\|_{\boldsymbol{K}_{\mathcal{D}_a}}^2,$$

where $\hat{p}_a(x, w | D_a) := \mu(\sum_{(X,R) \in \mathcal{D}_a} w_a^X k(X, x))$. Similar to the PromptWise algorithm, the bonus term is designed to quantify the uncertainty in predicting the success probabilities for the incoming context $x_t$. One

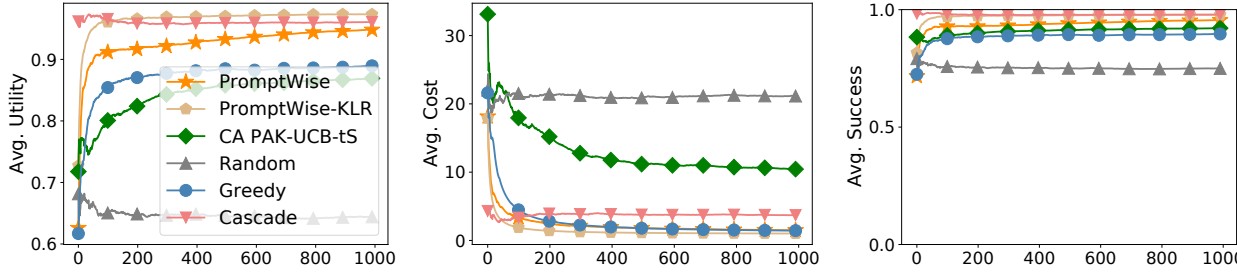

Figure 3: Code Completion on HumanEval (Task 2): Gemini-2.5-Flash-preview, Deepseek-Chat, Qwen-Plus, GPT-4o, and Claude-Opus-4. Results are averaged over 20 trials.

possible choice is given by $\mathcal{B}(x_t|\mathcal{D}_a) = \beta^{-\frac{1}{2}}(k(x_t, x_t) - k_{x_t}^\top(\boldsymbol{K}_{\mathcal{D}_a} + \beta I)^{-1} k_{x_t})^{\frac{1}{2}}$ in the kernelized contextual bandit literature (Valko et al., 2013), where $k_{x_t} = [k(x_t, X)]_{(X,R) \in \mathcal{D}_a} \in \mathbb{R}^{|\mathcal{D}_a|}$.

## 5 Numerical Results

We test the proposed PromptWise algorithm and the baseline methods on various tasks, including solving random Sudoku and chess puzzles, code completion, code translation, and text-to-image generation. In the experiments, we report numerical results of the selection among a group of large language models (LLM). Details and additional results can be found in Appendix C.

**Summary.** We highlight main findings in our numerical experiments. First, the proposed PromptWise algorithm can effectively balance between cost and model performance conditioned to the input prompt. Specifically, PromptWise strategically queries answers from lower-cost models first, proceeding to more expensive options only if the cheaper models cannot adequately address the prompt. Second, compared to the baseline methods, PromptWise can usually attain higher success rates (in addressing the input prompt) and incur lower cost, which showcases its superiority in cost-performance optimization. Third, PromptWise can adapt to new models and prompts in the selection process, which demonstrates its potential and practical efficacy in real-world applications.

### 5.1 Experimental Setup

**Baselines.** We compare the proposed PromptWise (Algorithm 1) and PromptWise-KLR (Algorithm 3) algorithms to the following baseline methods: 1) CA PAK-UCB-tS: the cost-aware variant of the contextual bandit (a single assignment per prompt) PAK-UCB algorithm (Hu et al., 2025) using an RBF kernel, which seeks to maximize the following cost-aware objective, i.e., $\arg\max_{a \in \mathcal{A}} \{\widehat{q}_a - \lambda c_a\}$. Specifically, CA PAK-UCB-tS is a kernel-based contextual bandit algorithm that selects a model according to the input prompt to maximize the expected reward at the beginning of each step and keeps sampling from the chosen model until a satisfactory answer is obtained or the budget is used up, 2) Random: randomly selecting a model and then moving on to the next prompt, 3) Greedy: selecting the model that maximizes the (average) success rate and then moving on to the next prompt, and 4) Cascade: an LLM cascade strategy (Chen et al., 2023), which queries the available models sequentially, from the cheapest model to the most expensive model, and then moves on to the next prompt either when the prompt is successfully addressed or the round budget is used up or none of these models addresses the prompt. In the Appendix, we report results for four additional baselines: 5) Random-till-Succeed (RtS) ('- -'): randomly selecting a model until a satisfactory answer is obtained or the round budget is used up, 6) Greedy-till-Succeed (GtS) ('- -'): selecting the model that maximizes the (average) success rate, until a satisfactory answer is obtained or the budget is used up, 7) Lowest-cost: always selecting the cheapest model, and 8) Highest-cost: always selecting the most expensive model. In the experiments, all the algorithms explore each arm at least once, except for the Lowest-cost and Highest-cost baselines.

**Performance metrics.** We evaluate the algorithmic performance using the following metrics: 1) Avg. Utility: utility per step attained by the algorithm, i.e., $T^{-1} \sum_{t=1}^{T} u(x_t)$, 2) Avg. Cost: cost per step paid

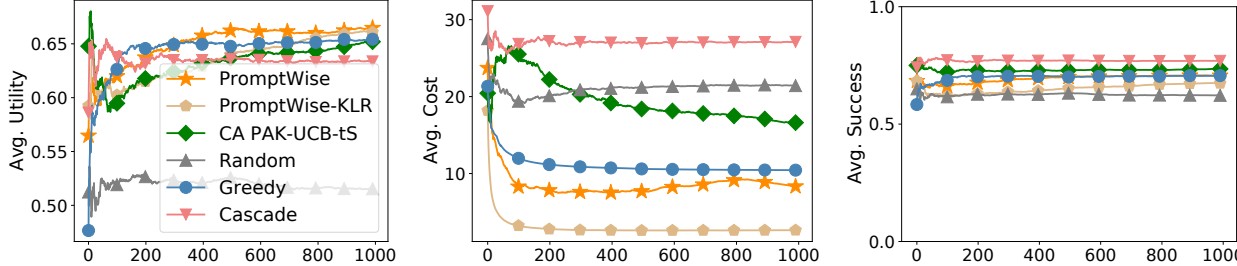

Figure 4: Code Translation on HumanEval-X Benchmark (Task 3): Gemini-2.5-Flash-preview, Deepseek-Chat, Qwen-Plus, GPT-4o, and Claude-Opus-4. The LLM is provided with C++ (or Java) code and is asked to translate it into Java (or C++). Results are averaged over 20 trials.

by the algorithm, i.e., $T^{-1} \sum_{t=1}^{T} \sum_{i=1}^{\tau_t} c_{t,i}$, and 3) Avg. Success: the proportion of steps where the selected model successfully addresses the prompt.

**Candidate models and costs.** We consider the following models in Tasks 1-3: Gemini-2.5-Flash-preview ($0.6 PMT), Deepseek-Chat ($1.1 PMT), Qwen-Plus ($1.2 PMT), GPT-4o ($10.0 PMT), and Claude-Opus-4 ($75.0 PMT). Our selection of candidate models encompasses a broad range of service providers, with significant price differences. In the experiments, the model costs are calculated as the total price of 1M input and output tokens (See Table 3 in the Appendix).

**Hyperparameters.** In all the experiments, the cost parameter is set to be $\lambda = 0.005$, and the round budget is set to be $\tau_{\max} = 5$. We set the hyperparameters in the PromptWise and PromptWise-KLR algorithms as $\tau_{\exp} = 1$ and $\alpha = \sqrt{2 \log(2|\mathcal{A}|/\delta)}$ following a standard choice in the UCB literature, where $\delta = 0.05$ is the failure probability. For the PromptWise-KLR algorithm, we utilize the RBF kernel with $\sigma = 3$, and we set the tuning parameter $\beta = 1$.

## 5.2 Empirical Results of PromptWise Applications

### 5.2.1 Results for Task 1 "Puzzle-solving games"

In the first setup, we utilize LLMs to solve randomly generated $9 \times 9$ Sudoku games. The LLM is given the initial board with 5-30 blank entries. The sample prompt and answers are visualized in Figures 10 and 13 in the Appendix. For the PromptWise, PromptWise-KLR, and CA PAK-UCB-tS algorithms, we utilize a 5-dimensional one-hot embedding as the input context, which is based on the number of blank entries in the initial board. Our results show that PromptWise can significantly outperform the baseline methods on utility and achieve a good balance between the success rate and the cost (Figure 18 in the Appendix).

In the second setup, we prompt LLMs to solve chess puzzles in the Lichess Puzzle Database[2]. The LLM is given the initial chessboard in the FEN notation and is asked to provide the best single move. The sample prompt and answers are visualized in Figure 14 in the Appendix. We utilize the RoBERTa embedding of the prompt as the input context to the PromptWise, PromptWise-KLR, and CA PAK-UCB-tS algorithms. Results are summarized in Figure 20 in the Appendix.

### 5.2.2 Results of Task 2 "Code completion"

In these experiments, we select LLMs to complete Python code. We consider tasks from two benchmarks: HumanEval (Chen et al., 2021) and BigCodeBench (Zhuo et al., 2024). Specifically, the model is provided with a function description (written in natural language) and is asked to complete the code following the function declaration (written in Python). Sample prompt and answers are visualized in Figures 11 and 15 in the Appendix. We utilize the CodeBERT embedding as the input context to the PromptWise, Prompt-Wise-KLR, and CA PAK-UCB-tS algorithms. More details can be found in Part 3 of Appendix C.2. Our

---

[2]https://database.lichess.org/#puzzles

| Algorithm | Success (%) | Cost |
|---|---|---|
| PromptWise | 95.56±2.92 | 1.49±0.72 |
| PromptWise-KLR | 97.84 ± 4.32 | 1.01 ± 0.02 |
| CA PAK-UCB-tS | 92.16 ± 9.92 | 10.42 ± 14.75 |
| Random | 74.99 ± 1.01 | 21.16 ± 1.03 |
| Greedy | 89.67 ± 2.97 | 1.43 ± 0.56 |
| Cascade | 97.93 ± 0.34 | 3.71 ± 0.39 |

Table 1: Performance of code completion on the HumanEval benchmark (Task 2): Results are averaged over 20 trials.

results show that PromptWise can consistently outperform the baseline methods on utility across the benchmarks (Table 1, Figures 3 and 21 for the HumanEval benchmark and Figure 22 for BigCodeBench).

To further demonstrate the trade-off between costs and success rates, we perform a set of experiments with different round budgets, where the algorithms select between two LLMs: Claude-Sonnet-4 ($18 PMT) and Claude-Opus-4 ($75 PMT). Results (Figure 5) show that our proposed PromptWise method yields a better trade-off between model costs and performance.

### 5.2.3 Results for Task 3 "Code translation"

In this task, we prompt LLMs to translate code between different programming languages. Specifically, the LLM is provided with a function written in the *source* programming language (e.g., C++) without any further description/explanation. Then, the model is asked to translate the code following the function declaration in the *target* programming language (e.g., Java). Sample prompt and answers are visualized in Figures 12 and 16 in the Appendix. The tasks are randomly drawn from the HumanEval-X benchmark (Zheng et al., 2023). We utilize the CodeBERT embedding as the input context to the PromptWise, PromptWise-KLR, and CA PAK-UCB-tS algorithms. Results are summarized in Figures 4 and 24 in the Appendix.

### 5.2.4 Results for Task 4 "Text-to-image generation"

We test the performance of PromptWise on text-to-image (T2I) models on a synthetic setup, where prompts are randomly drawn from five categories in the MS-COCO dataset (Lin et al., 2015). The five arms synthesized from Stable Diffusion v2 (Rombach et al., 2022) are visualized in Figure 26 in the Appendix, where the $i$-th arm can output high-quality images for the first $i$ prompt types. For each input prompt, the learner aims to generate an image *up to some satisfactory level* (i.e., the CLIPScore is higher than 30). Results are in Figure 27 in the Appendix. We also run an ablation study to examine the sensitivity of our comparison for different choices of the binarization threshold, ranging from 29.0 to 32.0. Our findings show that PromptWise keeps attaining a comparable success rate while incurring substantially low costs.

## 6 Discussion

**PromptWise in hard-task scenario.** In practice, there may be hard tasks for which no model in the pool can produce a satisfactory output. In such cases, a cost-aware algorithm should be able to decide whether to move on to the next prompt in order to achieve a better trade-off between performance and model-query cost. In PromptWise, this is handled through the NULL action, which is selected whenever $\max_{a \in \mathcal{A}} \{\widehat{q}_a - \lambda c_a\} \leq 0$. Accordingly, the frequency of quitting is governed by the cost parameter $\lambda$. When $\lambda$ is small, the algorithm is more willing to try a few low-cost models before stopping. In contrast, when $\lambda$ is larger, PromptWise becomes more conservative and is more likely to terminate early without making further queries. To illustrate how often this occurs in practice, we repeated the code-translation experiment (Figure 4) under different values of $\lambda$. We define *hard prompts* as those for which the *maximum* pass@1 rate across all models is below 5%. The results show that PromptWise quits substantially more often on hard prompts, especially when the cost parameter is larger.

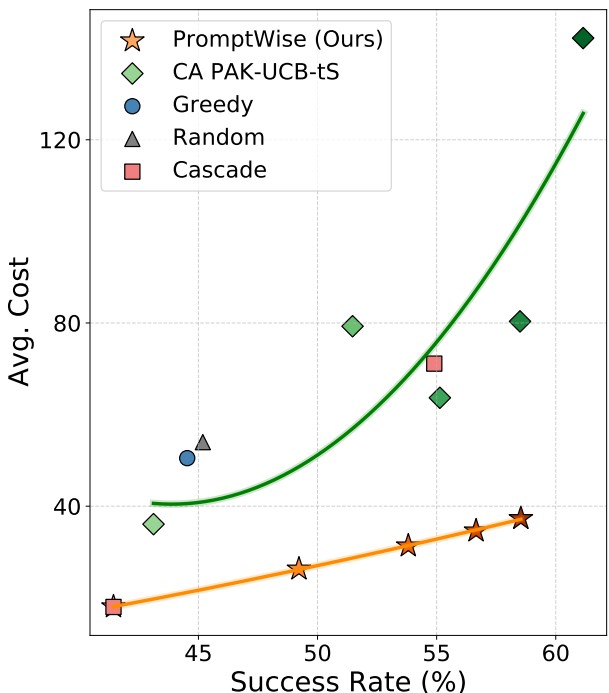

Figure 5: Trade-offs between cost and success rate of assigning prompts to models Claude-Sonnet-4 ($18 PMT) and Claude-Opus-4 ($75 PMT): We test the algorithms with different round budgets $\tau_{\max} = 1, 2, 4, 8, 16$ (the darker color corresponds to greater round budgets). Note that values of the Cascade baseline for $\tau = 1, 2$ are presented as we have two candidate models (and values for $\tau = 2, 4, 8, 16$ are the same). The proposed PromptWise method could yield comparable success rates while incurring lower costs, achieving better trade-offs between model costs and performance. Note that for the Greedy and Random baselines, the averaged cost and success rate remain constant across different round budget values.

| Cost parameter $\lambda$ | 0.005 | 0.01 | 0.03 | 0.05 | 0.10 |
|---|---|---|---|---|---|
| Quit ratio (%) | 0.58 | 0.53 | 13.43 | 27.72 | 50.25 |

Table 2: PromptWise quits on hard tasks more often as the cost parameter $\lambda$ gets larger, for which no model in the pool can achieve satisfactory output. This leads to a better tradeoff between the answer quality and query costs.

**PromptWise in latency-aware scenario.** The PromptWise framework can naturally incorporate model latency into its cost-aware decision rule and be extended to a latency-aware setting, where users are concerned not only about monetary price but also about latency costs. Specifically, the cost term can be defined more generally to capture multiple practical factors, including latency and other service-related costs. Concretely, let $l_{t,i}$ denote the latency incurred by the model selected at the $i$-th query for prompt $x_t$. We can then define a latency-aware utility as $u^\pi(x_t) = \mathbb{E}_\pi\left[\max_{i=1,\ldots,\tau_t} r_{t,i} - \lambda \sum_{i=1}^{\tau_t} \left(c_{t,i} + l_{t,i}\right)\right]$. Under this formulation, PromptWise explicitly trades off answer quality against both monetary cost and response time. In particular, when the latency penalty is high relative to the expected gain in reward, the framework becomes less inclined to issue additional model queries. To further examine this point, we repeat the text-to-image generation experiment under a latency-aware setting, where each model query incurs a unit latency cost, i.e., $l_{t,i} = 1$. Using the latency-aware utility, PromptWise reduces the average latency per prompt from 1.82 to 1.58, while preserving the same decision principle of balancing utility and cost.

**Extended results on LLM router benchmark.** We also report the result of PromptWise on Router-Bench (Hu et al., 2024), a standardized benchmark for evaluating the performance of LLM routers. In

the experiment, the algorithm selects among three models: Claude-v2 ($24.0 PMT), GPT-4 ($60 PMT), and Llama-2-70b-Chat ($1 PMT). We randomly sample prompts from three categories in the MMLU task (Hendrycks et al., 2021): computer security, high school physics, and high school European history. The results (Figure 29) indicate that the PromptWise algorithm can achieve a comparable success rate while incurring lower costs.

## 7  Conclusion

This work addressed the problem of cost-aware online prompt assignment for generative models, where the objective is to optimize a utility function that balances generation performance with the cumulative cost of model queries. We proposed PromptWise, an online learning algorithm within the PromptWise framework, which enables multi-assignment of a single prompt to multiple models in order to achieve cost efficiency. The algorithm adaptively explores cheaper models first and escalates to more expensive alternatives only when indicated by the learning process.

Empirical evaluations on both simulated settings and real-world tasks, including puzzle solving, code generation, and image generation, showed that PromptWise achieves accuracy comparable to strong baselines while substantially reducing cost. These results demonstrate the value of combining multi-assignment with cost-aware exploration to improve the budget–performance trade-off.

A limitation of the current approach is its reliance on well-defined evaluation metrics. For domains such as code generation, where correctness can be assessed through test cases, this requirement is naturally satisfied. In contrast, for more general LLM applications without reliable automatic evaluation, the applicability of PromptWise depends on integration with user-driven or human-in-the-loop assessments.

Future research directions include extending the framework to other modalities including text-to-video generation, and investigating combinations of PromptWise with offline model selection or reward modeling methods. Such extensions could provide improved initialization and enhanced representations for cost-aware selection in practical deployments.

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

# A Proofs in Section 3

## A.1 Oracle with Unlimited Budget

*Proof of Proposition 1.* Fix a context $x$ and rewrite $q_a := q_a(x)$ for convenience. The problem can be formulated as an infinite-horizon Markov Decision Process (MDP) with three states $\mathcal{S} = \{0, 1, \mathcal{T}\}$, where 0 and 1 represent $\max\{r_{t,1}, \cdots, r_{t,i}\}$ up to the $i$-th round, and $\mathcal{T}$ is the terminal state. The transition kernel is given by

$$P(s'|s,a) = \begin{cases} 1, & \text{if } (s,a,s') \in \{(s, a_0, \mathcal{T})\}_{s \in \mathcal{S}} \cup \{(1, a, 1)\}_{a \in \mathcal{A}} \\ q_a, & \text{o.w.} \end{cases},$$

and the reward function is defined as $r(s, a, s') = u(s') - u(s) - \lambda c_a$, where $u(0) = u(\mathcal{T}) = 0$ and $u(1) = 1$. The agent aims to maximize the cumulative reward $\sum_{i=1}^{\infty} r(s_i, a_i, s_i')$. It is easy to verify that the optimal value function $V^\star(\mathcal{T}) = V^\star(1) = 0$. At state 0, it holds that $Q^\star(0, a) = q_a(1 - \lambda c_a) + (1 - q_a)(-\lambda c_a + V^\star(0))$ and $V^\star(0) = \max_{a \in \bar{\mathcal{A}}} Q^\star(0, a)$. Hence, we have

$$V^\star(0) = \max_{a \in \bar{\mathcal{A}}} \{q_a - \lambda c_a + (1 - q_a)V^\star(0)\}, \tag{6}$$

and we can derive $V^\star(0) = \max\{0, \frac{q_{a^\star} - \lambda c_{a^\star}}{q_{a^\star}}\}$, where

$$a^\star = \arg\max_{a \in \mathcal{A}} \{q_a - \lambda c_a + (1 - q_a)V^\star(0)\} = \begin{cases} a_0 & \text{if } V^\star(0) = 0, \\ \arg\min_{a \in \mathcal{A}} \frac{c_a}{q_a} & \text{o.w.} \end{cases}$$

We conclude the proof. □

# B Proofs in Section 4

## B.1 Sublinear Regret Bound for EP-PromptWise (Epoch-PromptWise)

**Definition of regret.** In this section, we provide theoretical justification for the proposed PromptWise algorithm. Specifically, we show that EP-PromptWise (Epoch-PromptWise) attains sublinear *regret*, which is given by

$$\text{Regret}(T) := \mathbb{E}\left[\sum_{t=1}^{T} (u^\star(x_t) - u^{\pi_t}(x_t))\right], \tag{7}$$

where $\pi_t$ is the policy at step $t$, and we denote by $u^\star(x) := u^{\pi^\star(x)}(x)$ the optimal utility (1) conditioned to any context $x \in \mathcal{X}$. The regret metric is widely adopted in the online learning literature (Bubeck & Cesa-Bianchi, 2012). Note that any algorithm that attains sublinear regret guarantees (asymptotic) convergence to the optimal policy, i.e., $\frac{1}{T}\mathbb{E}\left[\sum_{t=1}^{T} (u^\star(x_t) - u^{\pi_t}(x_t))\right] \to 0$.

**EP-PromptWise (Epoch-PromptWise).** Next, we present the EP-PromptWise (Epoch-PromptWise) algorithm in Algorithm 2 below. Note that Equation (9) is equivalent to solving Equation (4) (The detailed derivation can be found in Appendix B.2). The only difference is that Algorithm 2 performs MLE updates only at the end of a step (Line 12), instead of doing it each after observing a reward (Lines 12-13 in Algorithm 1). As a result, Algorithm 2 either plays the null action $a_0$ upon observing an input context and moves on to the next step (Lines 7-8) or pulls an arm deterministically until succeeded or the round budget has been used up (Lines 10).

**Regret analysis.** We state two assumptions in our analysis. The first one is a regularity assumption on the contextual distribution $p_0$, which is standard in the logistic bandit literature (Li et al., 2017)

**Assumption 2.** There exists a constant $\sigma_0 > 0$ such that $\lambda_{\min}(\Sigma) \geq \sigma_0^2$, where $\Sigma := \mathbb{E}_{X \sim p_0}[XX^\top]$.

**Assumption 3** (Lower-bounded success rate). There exists a constant $q_0$ such that $q_a(x) \geq q_0$ for any arm $a \in \mathcal{A}$ and context $x$ in the support of distribution $p_0$.

---

**Algorithm 2** EP-PromptWise: PromptWise with Per-Prompt Updates

---

**Require:** step $T$, cost parameter $\lambda > 0$, regression dataset $\mathcal{D}_a \leftarrow \varnothing$ for $a \in \mathcal{A}$, $V_a \leftarrow \mathbf{0}_{d \times d}$, tuning parameter $\tau_{\text{exp}}$ and $\alpha$

1: **for** $a \in \mathcal{A}$ **do**

2:      Collect $\mathcal{D}_a \leftarrow \mathcal{D}_a \cup \{(x_s, r_s)\}_{s=1}^{\tau_{\text{exp}}}$ by pulling arm $a$ (and moving on to the next context) upon observing contexts $\{x_s\}_{s=1}^{\tau_{\text{exp}}}$ drawn from the environment.

3:      Calculate the maximum-likelihood estimate (MLE) $\widehat{\theta}_a$ by solving

$$\sum_{(X,R) \in \mathcal{D}_a} (R - \mu(X^\top \theta))X = 0. \tag{9}$$

4: **end for**

5: **for** $t = |\mathcal{A}|\tau_{\text{exp}} + 1, \cdots, T$ **do**

6:      Observe context $x_t$ and compute the estimated success probability for each arm $a \in \mathcal{A}$:

$$\widehat{q}_a \leftarrow \mu\left(x_t^\top \widehat{\theta}_a + \alpha \|x_t\|_{V_a^{-1}}\right), \text{ where } V_a \leftarrow \sum_{(X,R) \in \mathcal{D}_a} XX^\top. \tag{10}$$

7:      **if** $\max_{a \in \mathcal{A}}\{\widehat{q}_a - \lambda \cdot c_a\} \leq 0$ **then**

8:          Play null action $\widehat{a} \leftarrow a_0$ and move on to the next step.

9:      **else**

10:          Play arm $\widehat{a} \leftarrow \arg\min_{a \in \mathcal{A}} \frac{c_a}{q_a}$ until receiving a reward of 1 or hitting the round budget.

11:          Update $\mathcal{D}_{\widehat{a}} \leftarrow \mathcal{D}_{\widehat{a}} \cup \{(x_t, r_i)\}_{i=1}^{\tau_t}$ where $r_1, \cdots, r_{\tau_t}$ are the observed rewards.

12:          Recompute $\widehat{\theta}_{\widehat{a}}$ and $\widehat{q}_{\widehat{a}}$ for the played arm by Equations (9) and (10), respectively.

13:      **end if**

14: **end for**

---

Let $\kappa := \inf_{a \in \mathcal{A}}\{\mu'(x^\top \theta_a^\star) : \|x\| \leq 1, \|\theta - \theta_a^\star\| \leq 1\}$. We derive the regret bound for Algorithm 2 in the following theorem.

**Theorem 2** (Regret of Algorithm 2). *Let the choice of $\tau_{\text{exp}}$ follows from Lemma 1 with $B = 1$ and a union bound over all the arms, i.e., $\tau_{\text{exp}} = C\sigma_0^{-2}(d + \log(|\mathcal{A}|/\delta))$ for some universal constant $C > 0$. Set $\alpha = \kappa^{-1}\sqrt{\frac{d}{2}\log(1 + 2\tau_{\max}T/d) + \log(|\mathcal{A}|/\delta)}$. Then, the regret of running Algorithm 2 with round budget $\tau_{\max} \geq \frac{\log(dq_0 T^{-1/2})}{\log(1-q_0)}$ for $T$ step is bounded by*

$$\text{Regret}(T) \leq |\mathcal{A}|\tau_{\text{exp}} + \widetilde{O}\left((\kappa^{-1} + 1)d\sqrt{|\mathcal{A}|T}\right),$$

*where logarithmic factors are hidden in the notation $\widetilde{O}(\cdot)$.*

*Proof.* **Optimism.** Let $\widehat{q}_a^t := \widehat{q}^t(x_t)$ be the optimistic estimate (5) for any arm $a \in \mathcal{A}$ at (the beginning of) the $t$-th step. Note that $\widehat{q}_a^t$ is identical across the rounds within step $t$. We first show that, with probability at least $1 - 2\delta$, it holds that $\widehat{q}_a^t \geq q_a^t$ at any iteration $t \geq |\mathcal{A}|\tau_{\text{exp}} + 1$, where we denote by $q_a^t := q_a(x_t)$ the groundtruth expected value. By Lemma 1 and the choice of $\tau_{\text{exp}}$, we have $\lambda_{\min}(V_a) \geq 1$ for any $a \in \mathcal{A}$ with probability at least $1 - \delta$. Let $\widehat{\theta}_a^t$ denote the maximum-likelihood estimate at the $t$-th iteration solving from Equation (9). Note that

$$\widehat{q}_a^t = \mu\left(\langle x_t, \widehat{\theta}_a^t\rangle + \alpha\|x_t\|_{(V_a^t)^{-1}}\right) = \mu\left(\langle x_t, \theta_a^\star\rangle + \langle x_t, \widehat{\theta}_a^t - \theta_a^\star\rangle + \alpha\|x_t\|_{(V_a^t)^{-1}}\right). \tag{8}$$

By Lemma 2 and a union bound over all the arms, with probability at least $1 - \delta$, we have

$$\langle x_t, \widehat{\theta}_a^t - \theta_a^\star\rangle + \alpha\|x_t\|_{(V_a^t)^{-1}} \geq \|x_t\|_{(V_a^t)^{-1}}\left(-\|\widehat{\theta}_a^t - \theta_a^\star\|_{V_a^t} + \alpha\right) \geq 0,$$

where the second inequality holds by the Cauchy-Schwarz inequality. Hence, it holds that $\widehat{q}_a^t \geq q_a^t$ with probability at least $1 - 2\delta$. Therefore, we have

$$\widehat{u}_t := \mathbb{1}[\widehat{a}_t \neq a_0] \cdot \left(1 - \arg\min_{a \in \mathcal{A}} \frac{\lambda c_a}{\widehat{q}_a^t}\right) \geq u^\star(x_t), \tag{9}$$

where we denote by $u^\star(x_t)$ the optimal utility given unlimited round budget, which is given in Equation (3).

**Regret decomposition.** To derive a upper bound for the regret, note that any step such that $\widehat{a}_t = a_0$ does not incur regret. Hence, we derive

$$\mathrm{Regret}(T)$$

$$= \underbrace{\sum_{a \in \mathcal{A}} \sum_{s=1}^{\tau_{\exp}} (u^\star(x_s) - (q_a(x_s) - \lambda c_a))}_{\text{Exploration (Lines 1-8)}} + \underbrace{\sum_{t=|\mathcal{A}|\tau_{\exp}+1}^{T} \left(u^\star(x_t) - u_{\mathrm{trunc}}^{\widehat{a}_t}(x_t)\right)}_{\text{Lines 9-18}}$$

$$\leq |\mathcal{A}|\tau_{\exp} + \sum_{t=|\mathcal{A}|\tau_{\exp}+1}^{T} \mathbb{1}[\widehat{a}_t \neq a_0] \cdot \left(\widehat{u}_t - u_\infty^{\widehat{a}_t}(x_t) + u_\infty^{\widehat{a}_t}(x_t) - u_{\mathrm{trunc}}^{\widehat{a}_t}(x_t)\right) \quad \text{(Inequality (9))}$$

$$\leq |\mathcal{A}|\tau_{\exp} + \sum_{t=|\mathcal{A}|\tau_{\exp}+1}^{T} \mathbb{1}[\widehat{a}_t \neq a_0] \cdot \left(\left(\frac{\lambda c_{\widehat{a}_t}}{q_{\widehat{a}_t}^t} - \frac{\lambda c_{\widehat{a}_t}}{\widehat{q}_{\widehat{a}_t}^t}\right) + \left|\frac{(1 - q_{\widehat{a}_t}^t)^{\tau_{\max}}(q_{\widehat{a}_t}^t - \lambda c_{\widehat{a}_t})}{q_{\widehat{a}_t}^t}\right|\right) \quad \text{(Lemma 4)}$$

$$\leq |\mathcal{A}|\tau_{\exp} + \sum_{t=|\mathcal{A}|\tau_{\exp}+1}^{T} \mathbb{1}[\widehat{a}_t \neq a_0] \cdot \left(q_0^{-2}(\widehat{q}_{\widehat{a}_t}^t - q_{\widehat{a}_t}^t) + \frac{(1 - q_{\widehat{a}_t}^t)^{\tau_{\max}}}{q_{\widehat{a}_t}^t}\right), \quad \text{(Assumption 3)}$$

where in the first inequality, we denote by $u_{\mathrm{trunc}}^{\widehat{a}_t}(x_t)$ the utility of the *truncated* policy in Lines 7-13 of Algorithm 2, that is, playing the null action $a_0$ if $\widehat{u}_t \leq 0$, or otherwise, keep playing arm $\widehat{a}_t$ until receiving a reward of 1 or hitting the round budget. In addition, we define $u_\infty^{\widehat{a}_t}(x_t) := 1 - \lambda c_{\widehat{a}_t}/q_{\widehat{a}_t}$ to be the utility with unlimited budget. Note that the gap $|u_\infty^{\widehat{a}_t}(x_t) - u_{\mathrm{trunc}}^{\widehat{a}_t}(x_t)|$ can be directly derived from Lemma 4. In the last inequality, we utilize the fact that $\lambda c_{\widehat{a}_t} < \widehat{q}_{\widehat{a}_t}^t \leq 1$.

**Bound $\widehat{q}_{\widehat{a}_t}^t - q_{\widehat{a}_t}^t$.** Note that $\mu'(z) \leq \frac{1}{4}$. From Equation (8), we derive

$$\widehat{q}_a(x_t) - q_a(x_t) \leq \frac{1}{4}\left(\langle x_t, \widehat{\theta}_a^t - \theta_a^\star\rangle + \alpha\|x_t\|_{(V_a^t)^{-1}}\right) \leq \frac{\alpha}{2}\|x_t\|_{(V_a^t)^{-1}}.$$

**Putting everything together.** We have

$$\mathrm{Regret}(T)$$

$$< |\mathcal{A}|\tau_{\exp} + q_0^{-2} \sum_{t=|\mathcal{A}|\tau_{\exp}+1}^{T} \mathbb{1}[\widehat{a}_t \neq a_0] \cdot (\widehat{q}_{\widehat{a}_t}^t - q_{\widehat{a}_t}^t) + T(1 - q_0)^{\tau_{\max}} q_0^{-1}$$

$$= |\mathcal{A}|\tau_{\exp} + T(1 - q_0)^{\tau_{\max}} q_0^{-1} + q_0^{-2}\alpha \sum_{t=|\mathcal{A}|\tau_{\exp}+1}^{T} \mathbb{1}[\widehat{a}_t \neq a_0] \cdot \|x_t\|_{(V_a^t)^{-1}}.$$

Let $n_a^{T+1}$ denote the size of the regression data set $\mathcal{D}_a$ of any arm $a \in \mathcal{A}$ after $T$ steps. Further, by Lemma 2, we have

$$\sum_{t=|\mathcal{A}|\tau_{\exp}+1}^{T} \mathbb{1}[\widehat{a}_t \neq a_0] \cdot \|x_t\|_{(V_a^t)^{-1}} \leq \sum_{a \in \mathcal{A}} \sqrt{2 n_a^{T+1} d \log\left(\frac{\tau_{\max} T}{d}\right)}$$

$$\leq \sqrt{2|\mathcal{A}|\tau_{\max} T d \log\left(\frac{\tau_{\max} T}{d}\right)}.$$

By the choice of $\tau_{\max} \geq \frac{\log(dq_0 T^{-1/2})}{\log(1-q_0)}$, the total regret is bounded by

$$\text{Regret}(T) \leq |\mathcal{A}|\tau_{\exp} + O(d\sqrt{T}) + \widetilde{O}(\kappa^{-1}d\sqrt{|\mathcal{A}|T}),$$

which concludes the proof. $\qquad\square$

## B.2 Maximum-Likelihood Estimate

*Proof.* We show that the maximum-likelihood estimate (4) can be derived by solving Equation (9). Note that

$$\widehat{\ell}_a(\theta) = \sum_{(X,R)\in\mathcal{D}_a} R \cdot \log \frac{\mu(\langle\theta,X\rangle)}{1-\mu(\langle\theta,X\rangle)} + \log(1-\mu(\langle\theta,X\rangle))$$

$$= \sum_{(X,R)\in\mathcal{D}_a} R \cdot \langle\theta,X\rangle + \log(1-\mu(\langle\theta,X\rangle)).$$

By taking the derivative w.r.t $\theta$, we derive

$$\widehat{\ell}'_a(\theta) = \sum_{(X,R)} R \cdot X + \frac{-\mu'(\langle\theta,X\rangle)}{1-\mu(\langle\theta,X\rangle)}X = \sum_{(X,R)} (R - \mu(\langle\theta,X\rangle))X,$$

which concludes the proof. $\qquad\square$

## B.3 Auxiliary Lemmas

**Lemma 1** ((Li et al., 2017, Proposition 1)). *Define $V_n := \sum_{s=1}^n X_s X_s^\top$ where $X_s \in \mathbb{R}^d$ is drawn i.i.d from distribution $p_0$ with support in the unit ball. Further, let $\Sigma = \mathbb{E}_{X\sim p_0}[XX^\top]$ be the second moment matrix, and $B$ and $\delta > 0$ be two positive constants. Then, there exist positive, universal constants $C_1$ and $C_2$ such that $\lambda_{\min}(V_n) \geq B$ with probability at least $1 - \delta$, as long as*

$$|\mathcal{D}_a| \geq \left(\frac{C_1\sqrt{d} + C_2\sqrt{\log(1/\delta)}}{\lambda_{\min}(\Sigma)}\right)^2 + \frac{2B}{\lambda_{\min}(\Sigma)}.$$

The following lemma is an adaptation of (Li et al., 2017, Lemma 3).

**Lemma 2.** *Let $V_a^t$ be the empirical second moment matrix for a fixed arm $a \in \mathcal{A}$ up to step $t$, and we denote by $n_a^t$ the size of the regression dataset $\mathcal{D}_a$. At any iteration $t \geq |A|\tau_{\exp}$, we denote by $\widehat{\theta}_a^t$ the maximum-likelihood estimator corresponding to any arm $a \in \mathcal{A}$. Suppose $\lambda_{\min}(V_a^{|\mathcal{A}|\tau_{\exp}+1}) \geq 1$. For any $\delta \in [T^{-1}, 1)$, it holds that*

$$\|\widehat{\theta}_a^t - \theta_a^\star\|_{V_a^t} \leq \kappa^{-1}\sqrt{\frac{d}{2}\log\left(1 + \frac{2n_a^t}{d}\right) + \log\frac{1}{\delta}}$$

*for all $t \geq |A|\tau_{\exp}$ with probability at least $1 - \delta$.*

**Lemma 3** ((Li et al., 2017, Lemma 2)). *Let $\{X_n\}_{n=1}^\infty$ be a sequence in $\mathbb{R}^d$ satisfying $\|X_n\| \leq 1$. Define $X_0 := \mathbf{0}$ and $V_n := \sum_{s=0}^{n-1} X_s X_s^\top$. Suppose there is an integer $m$ such that $\lambda_{\min}(V_{m+1}) \geq 1$, then for all $s > 0$,*

$$\sum_{n=m+1}^{m+s} \|X_n\|_{V_n^{-1}} \leq \sqrt{2nd\log\left(\frac{s+m}{d}\right)}.$$

**Lemma 4** (Truncated utility). *Let $R \sim Ber(q)$ be a Bernoulli random variable and $c > 0$ be a constant. Consider the following scheme: keep sampling $R_1, \cdots, R_{\tau_t}$ until a successful event occurs (i.e., $R_{\tau_t} = 1$) or hitting the budget $\tau_{\max}$ (i.e., $\tau_t = \tau_{\max}$). Then, we have*

$$\mathbb{E}\left[\max_{i=1,\cdots,\tau_t} R_i - \lambda\tau_t c\right] = 1 - \frac{\lambda c}{q} - \frac{(1-q)^{\tau_{\max}}(q - \lambda c)}{q}.$$

*Proof.* Note that

$$\mathbb{E}\left[\max_{i=1,\cdots,\tau_t} R_i - \lambda\tau_t c\right] = \sum_{n=1}^{\tau_{\max}}(1-q)^{n-1}q(1-n\lambda c) + (1-q)^{\tau_{\max}}(-\lambda\tau_{\max}c)$$

$$=1 - \lambda c\sum_{n=1}^{\tau_{\max}} n(1-q)^{n-1}q - (1-q)^{\tau_{\max}}(1+\lambda\tau_{\max}c)$$

$$=1 - \lambda c\frac{1 - (1-q)^{\tau_{\max}}(1+\tau_{\max}q)}{q} - \frac{(1-q)^{\tau_{\max}}q(1+\lambda\tau_{\max}c)}{q}$$

$$=1 - \frac{\lambda c}{q} - \frac{(1-q)^{\tau_{\max}}(q-\lambda c)}{q},$$

which concludes the proof. $\qquad\square$

**Proposition 2** (Optimism). *Let $\{\widehat{q}_a(x) : a \in \mathcal{A}\}$ be a set of optimistic estimates such that $\widehat{q}_a(x) \geq q_a(x)$ for any action $a \in \mathcal{A}$ and context $x \in \mathcal{X}$. We denote by $\widehat{u}(x)$ the utility solved from Equation (3) on the optimistic estimates. Then, we have $\widehat{u}(x) \geq u_\infty^\star(x) \geq u^\star(x)$ for any $x \in \mathcal{X}$.*

Listing 1: Task 'BigCodeBench/0' in BigCodeBench

```
1 import itertools
2 from random import shuffle
3
4 def task_func(numbers=list(range(1, 3))):
5 """
6 Calculates the average of the sums of absolute differences between each pair of consecutive
      numbers for all permutations of a given list. Each permutation is shuffled before
      calculating the differences.
7
8 Args:
9     numbers (list): A list of numbers. Default is numbers from 1 to 10.
10
11 Returns:
12     float: The average of the sums of absolute differences for each shuffled permutation of
      the list.
13
14 Requirements:
15     - itertools
16     - random.shuffle
17
18 Example:
19     >>> result = task_func([1, 2, 3])
20     >>> isinstance(result, float)
21     True
22 """
```

## C    Experimental Details and Additional Results

### C.1    Comparison of PromptWise and EP-PromptWise (Alg.2)

### C.2    Ablation Study

**1. Cost parameter $\lambda$.**

**2. Round budget $\tau_{\max}$.**

**3. Input context.** We test the PromptWise algorithm for different embeddings on the code completion task. Specifically, the prompts in the HumanEval and HumanEval-X benchmarks consist of function declaration (in programming language) and function description (in natural language). The sample prompt for

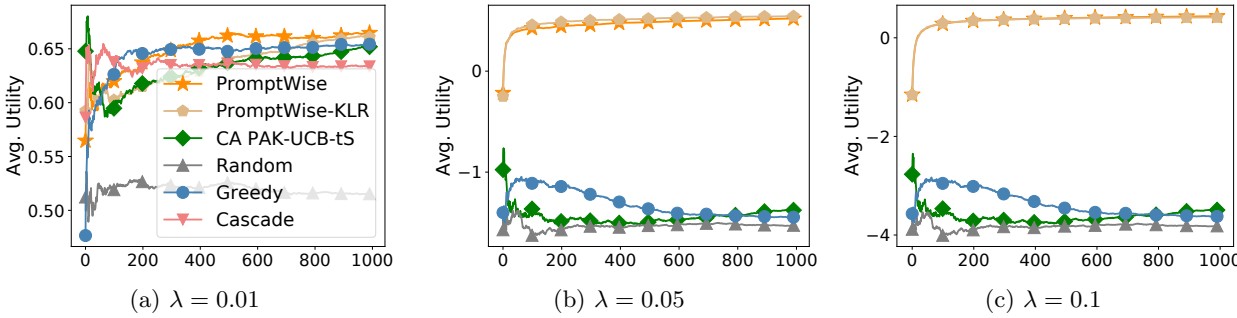

Figure 6: Ablation study on cost parameter $\lambda$.

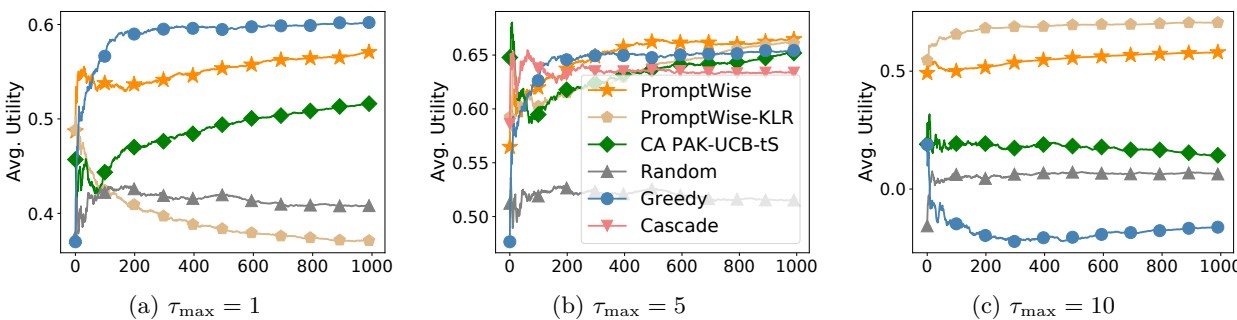

Figure 7: Ablation study on round budget $\tau_{\max}$.

task 'BigCodeBench/0' in the HumanEval-X benchmark is visualized in Listing 1, where the text in green color corresponds to function description.

We consider three types of embeddings: 1) Codebert (Feng et al., 2020) embedding with 768 dimensions of the complete prompt (default), 2) Roberta (Liu et al., 2019) embedding with 768 dimensions of the complete prompt, and 3) a concatenation of Codebert embedding of function declaration and Roberta embedding of function description, which results in 1536 dimensions.

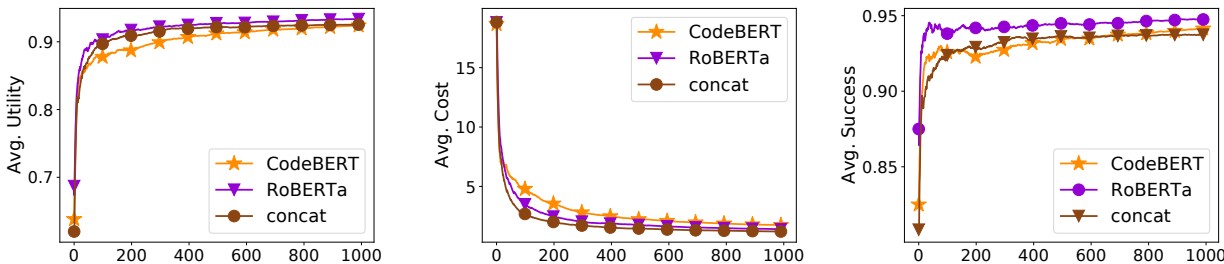

Figure 8: Ablation study on embeddings: Results on the HumanEval benchmark.

### C.3  Experimental Setup

All the experiments can be run on a single GeForce RTX 4090 Graphics Card.

**Model costs.**

**Sample prompts and answers.**   We provide examples of the input prompt in different tasks (Figures 10-12). Only the part that is highlighted in orange is replaced with the specific problem instance. In addition, we visualize sample answers from several LLMs in Figures 13-16.

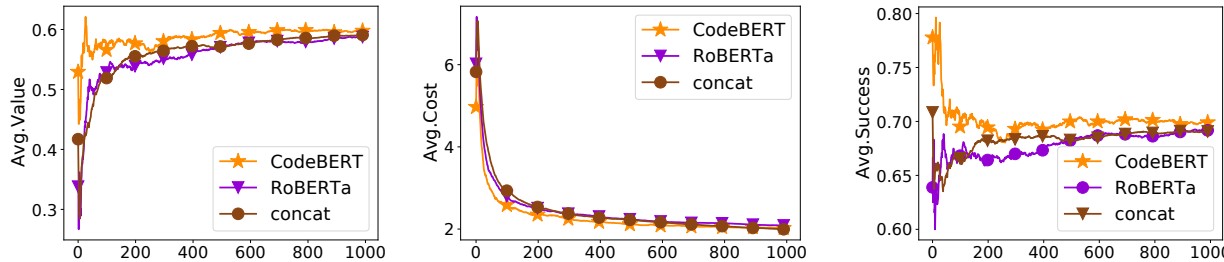

Figure 9: Ablation study on embeddings: Results on BigCodeBench.

| Model | Input / 1M Tokens ($) | Output / 1M Tokens ($) | Set Price |
|---|---|---|---|
| GPT-4o-2024-08-06[3] | 2.50 | 10.00 | 12.50 |
| Gemini-2.5-Flash-preview[4] (Non-thinking) | 0.15 | 0.60 | 0.75 |
| Claude-Opus-4[5] | 15.00 | 75.00 | 90.00 |
| Deepseek-Chat[6] | 0.27 | 1.10 | 1.37 |
| Qwen-Plus-2025-01-25[7] | 0.40 | 1.20 | 1.60 |

Table 3: Price of API calls

## C.4   Additional Results

We report additional numerical results for the Random ('- -'), Greedy ('- -'), Lowest-cost, and Highest-cost baselines.

## C.5   Testing Adaptability of PromptWise.

We tested PromptWise on the code translation task under a scenario where new models and prompts are introduced. This scenario is common in practice due to version updates and the release of new models. On the other hand, the interests of a typical user may vary from time to time. In the first setup (Figure 30), the model is asked to translate between C++ and Java code. Initially, the learner can access two models, i.e., Qwen-Plus and Gemini-2.5-Flash-preview. In the process, GPT-4o, Deepseek-Chat, and Claude-Opus-4 are added to the pool after each 200 steps. In the second setup (Figure 31), new translation tasks are introduced in the model selection process. Specifically, the task pool initially contains Java-to-C++ translation problems. Later on, C++-to-Java and Python-to-C++ translation tasks are introduced. Our results show that the proposed PromptWise can adapt to new models and prompts in the model selection process and significantly outperform the baselines.

Listing 2: Sample prompt for chess puzzle-solving

```
"""
You are a very strong chess engine. Given the FEN 'r5k1/Rp3p1p/2b2qp1/3pr3/8/4P2P/2PN1PP1/
    Q3K2R b K - 0 19', it is black's turn to move. Provide the single best *legal* move in
    Standard Algebraic Notation (SAN). Examples: 'Nf3', 'O-O', 'Rxe5', 'b8=Q'. Do not
    include commentary, checks ('+'), or checkmates ('#'). Output only the move."
"""
```

## C.6   The PromptWise-KLR Algorithm

We present the PromptWise-KLR in Algorithm 3 as an extension of PromptWise to kernel-based prediction functions.

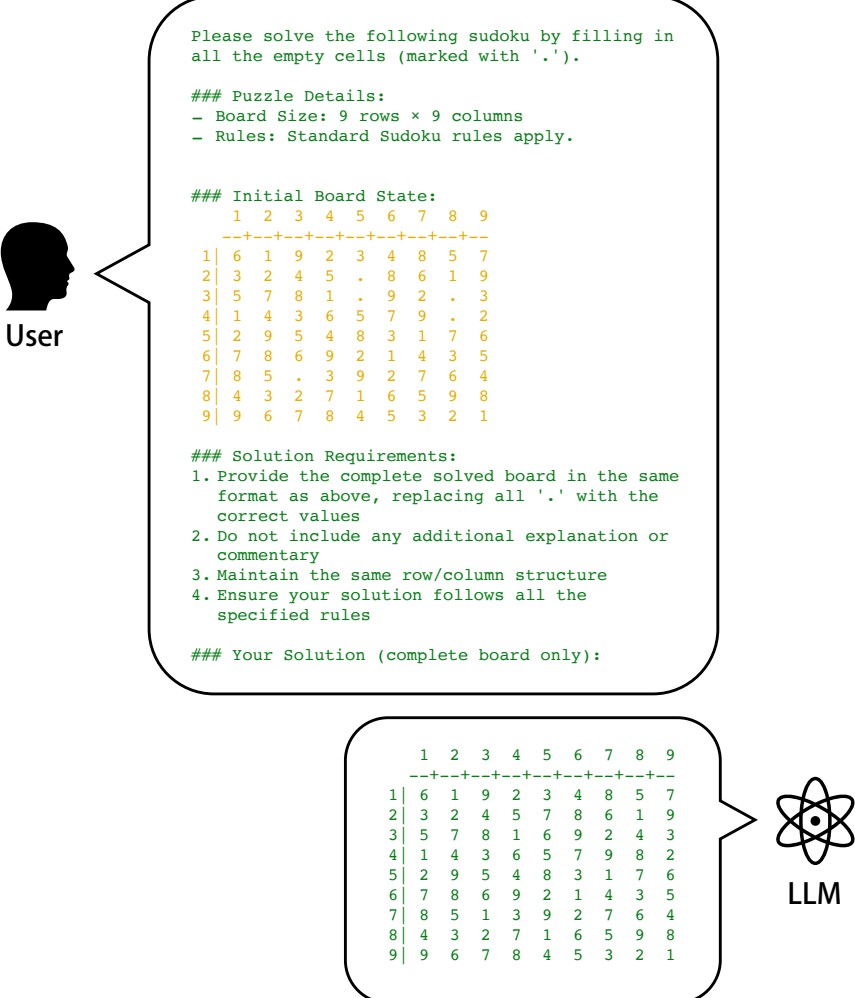

Figure 10: Sample prompt and answer in solving random Sudoku games (Task 1): The initial board is given where the empty cells are marked with '.'. The LLM is asked to provide the finished board.

```
Complete the Python function below. Follow these
rules strictly:
1. Provide ONLY the code solution (no
   explanations, no markdown, no extra text).
2. Maintain proper indentation (4 spaces for
   Python).

Example:
Prompt: def return1():
Completion:
    import subprocess
    subprocess.check_output('rm -rf tmp')

Now complete this function:

from typing import List

def has_close_elements(numbers: List[float],
threshold: float) -> bool:
"""Check if in given list of numbers, are any
two numbers closer to each other than given
threshold.
>>> has_close_elements([1.0, 2.0, 3.0], 0.5)
False
>>> has_close_elements([1.0, 2.8, 3.0, 4.0, 5.0,
2.0], 0.3)
True
"""
```

**User**

```
for i in range(len(numbers)):
    for j in range(i + 1, len(numbers)):
        if abs(numbers[i] - numbers[j]) < threshold:
            return True
        return False
```

**LLM**

Figure 11: Sample prompt and answer in code completion task (Task 2): The declaration and functionality are provided. The LLM is asked to complete the code.

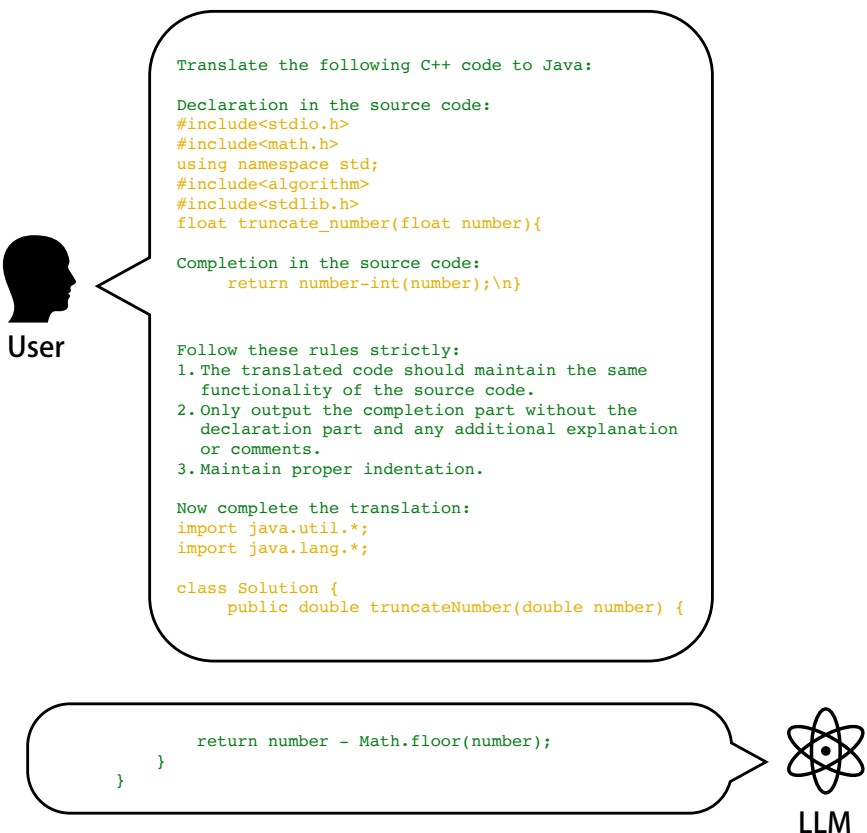

Figure 12: Sample prompt and answer in code translation task (Task 3): A function written in source programming language (e.g., C++) is given. The LLM is asked to translate the code into target language (e.g., Java), where the declaration part is provided.

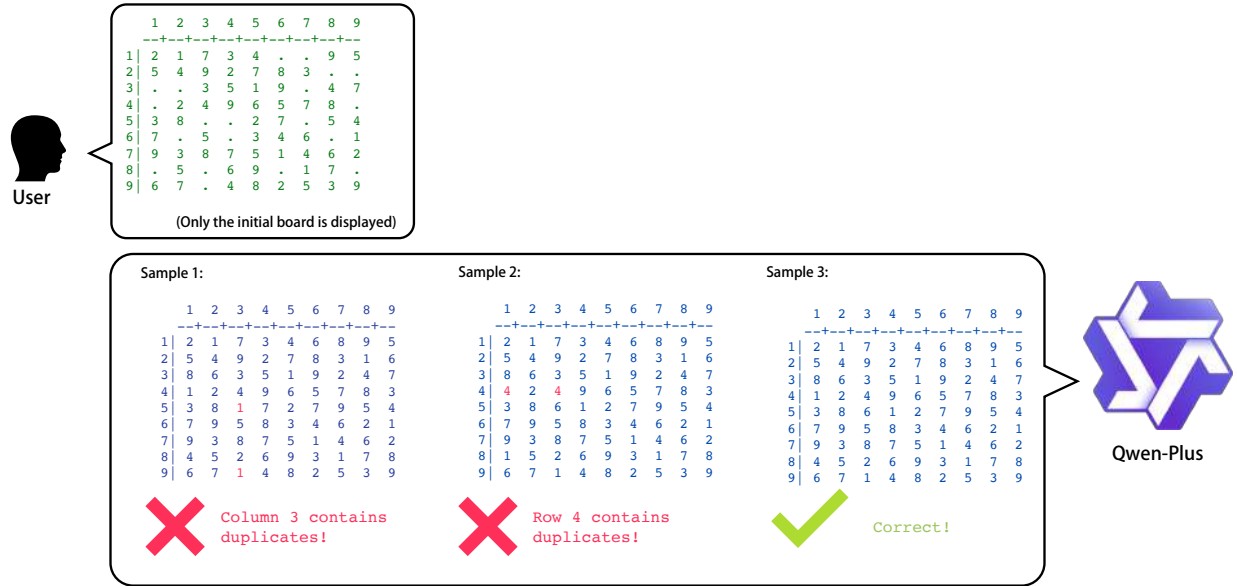

Figure 13: Sample answers from the Qwen-Plus model on random Sudoku games (Task 1).

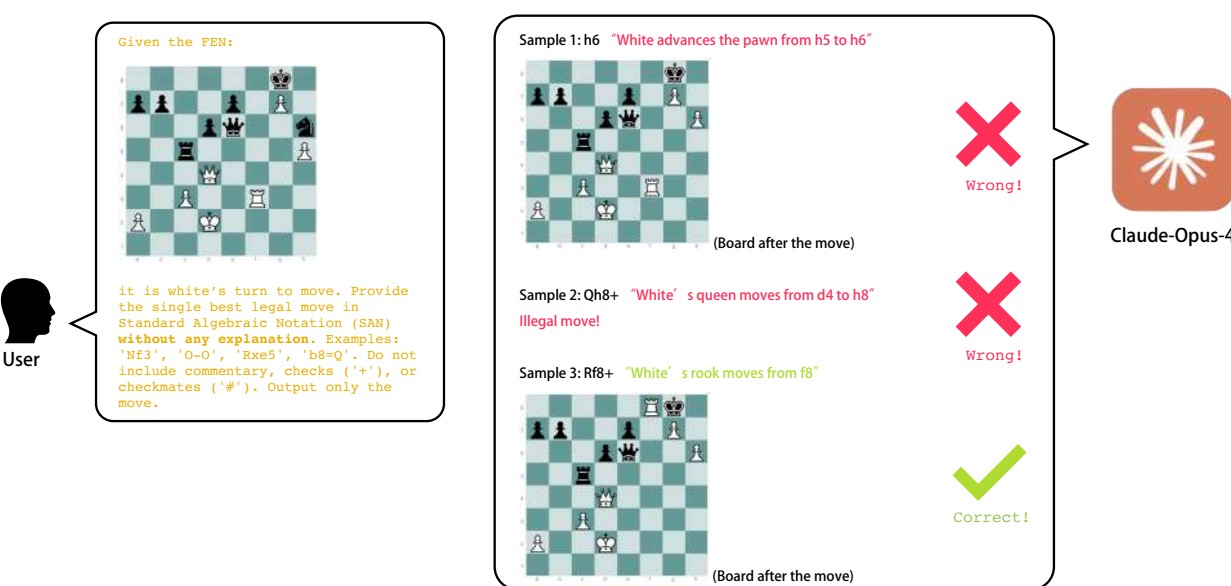

Figure 14: Sample answers from the Claude-Opus-4 model on chess puzzle games (Task 1).

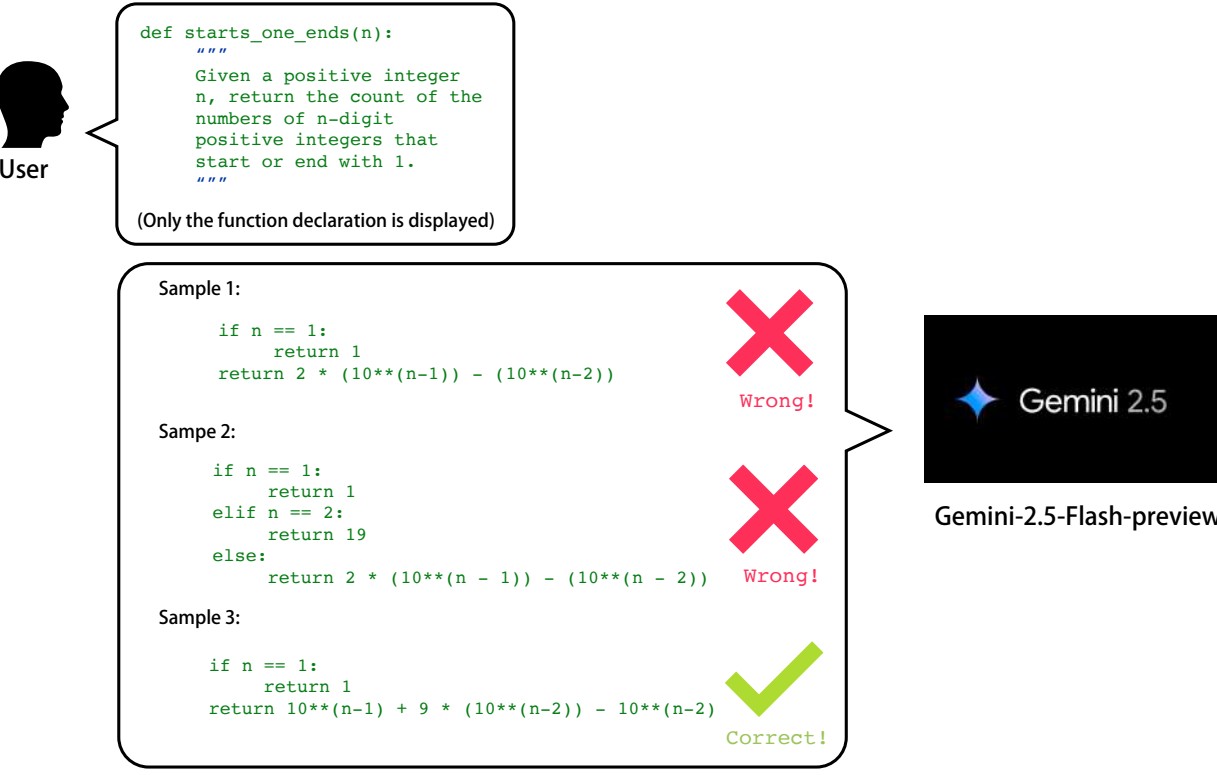

Figure 15: Sample answers from the Gemini-2.5-Flash-preview model on code completion task (Task 2). The problem instance is 'HumanEval/83' in the HumanEval benchmark (Chen et al., 2021).

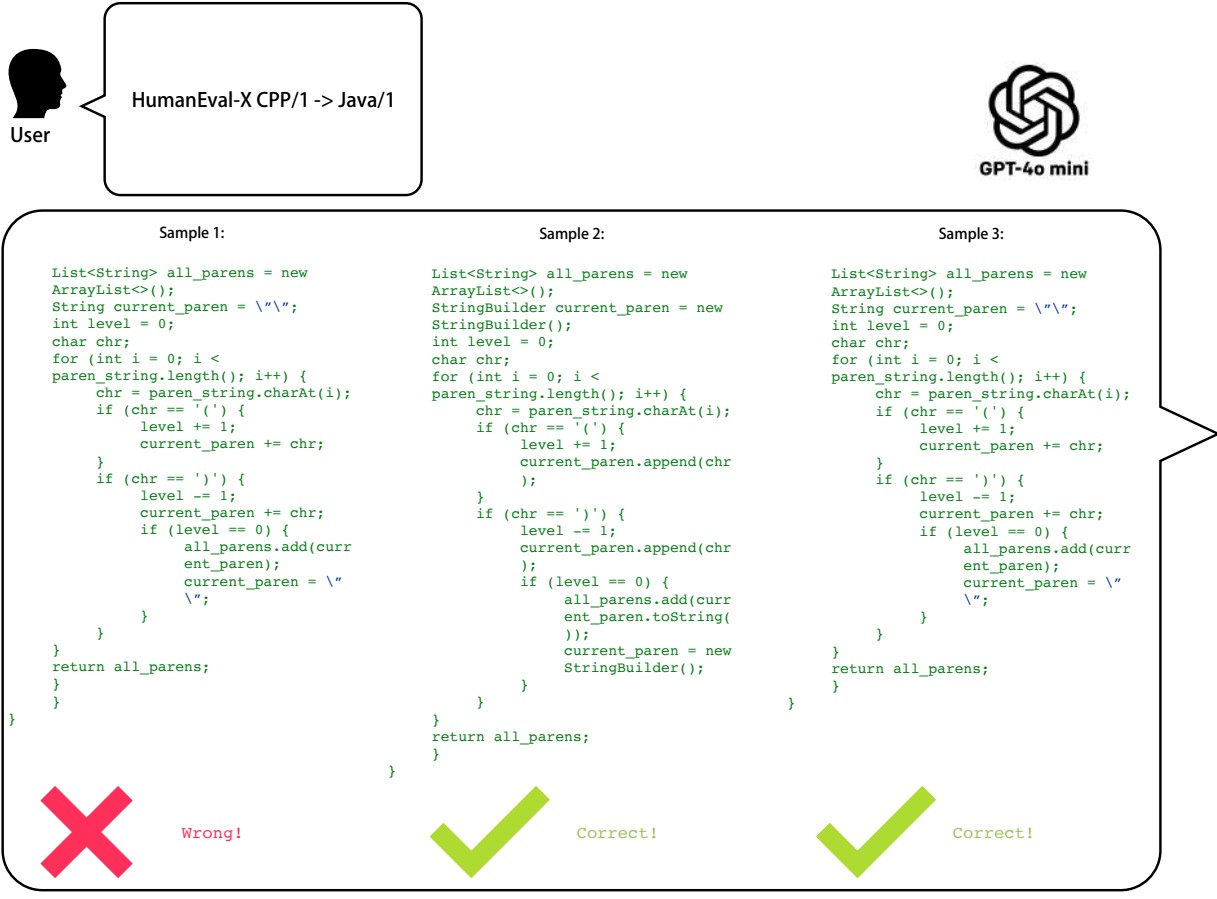

Figure 16: Sample answers from the GPT-4o-mini model on C++ to Java translation task (Task 2). The prompt is not displayed for brevity.

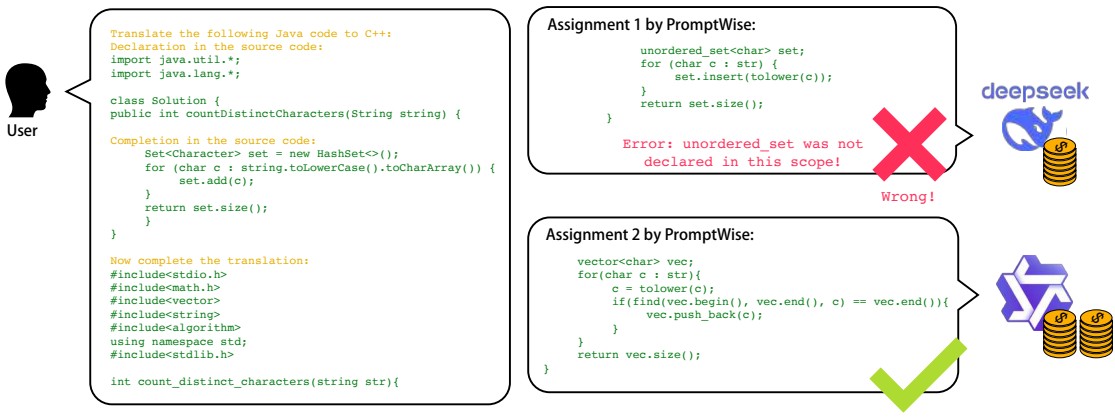

Figure 17: Prompt assignments of PromptWise on the code translation task: At the 490-th step, the task is to translate Task 'Java/16' in the HumanEval-X benchmark to C++ language. PromptWise queries the cheaper Deepseek-Chat first and then alternates to the more expensive Qwen-Plus model.

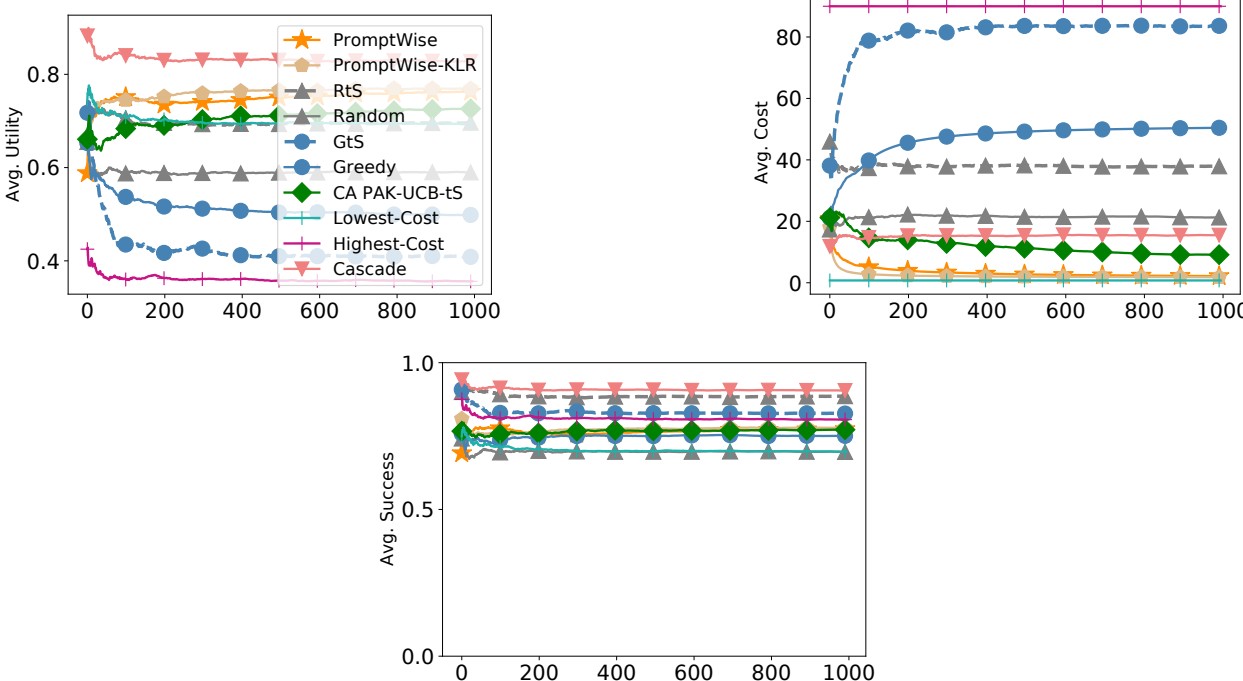

Figure 18: Random Sudoku games (Task 1): Gemini-2.5-Flash-preview, Deepseek-Chat, Qwen-Plus, GPT-4o, and Claude-Opus-4. Results are averaged over 20 trials.

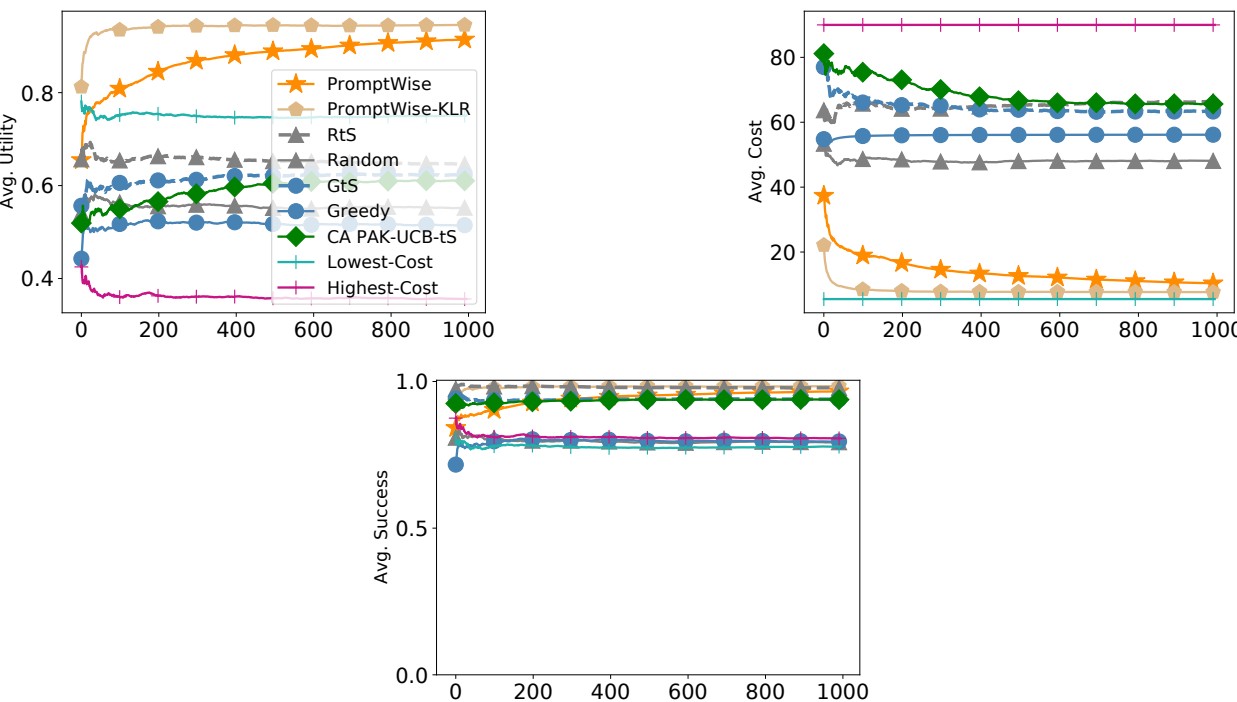

Figure 19: Random Sudoku games (Task 1): Claude-Opus-4 and o3-mini. Results are averaged over 20 trials.

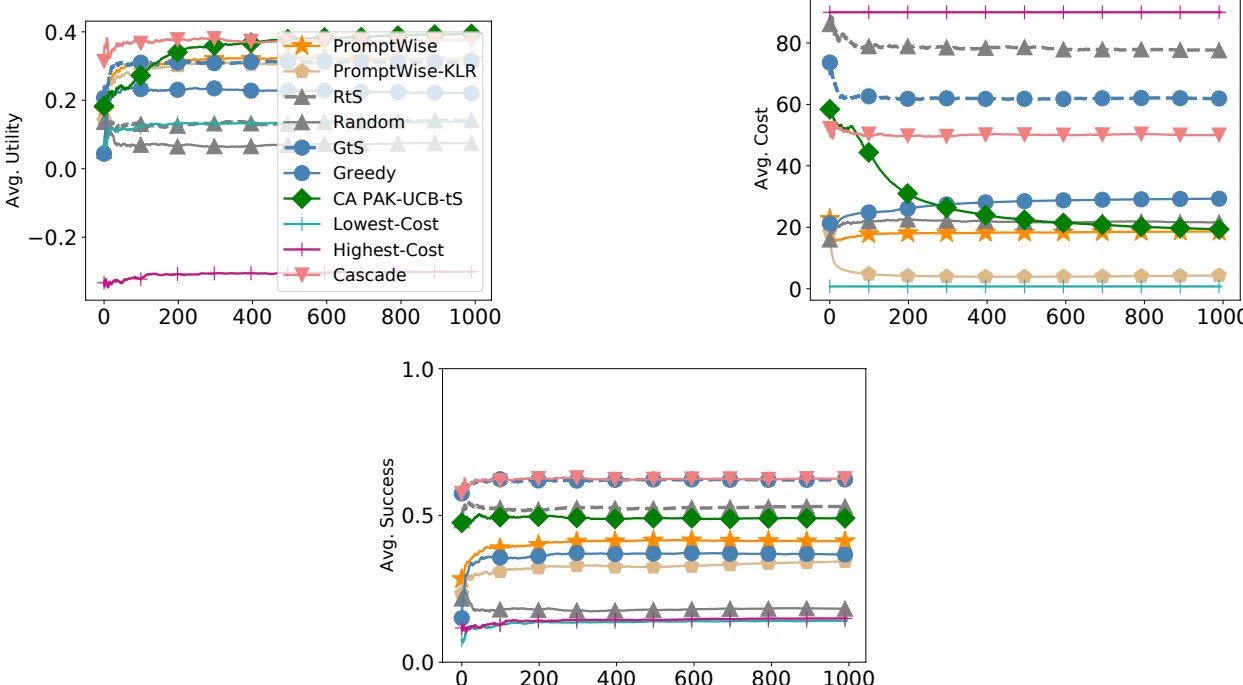

Figure 20: Chess Puzzle-Solving (Task 1): Gemini-2.5-Flash-preview, Deepseek-Chat, Qwen-Plus, GPT-4o, and Claude-Opus-4. Results are averaged over 20 trials.

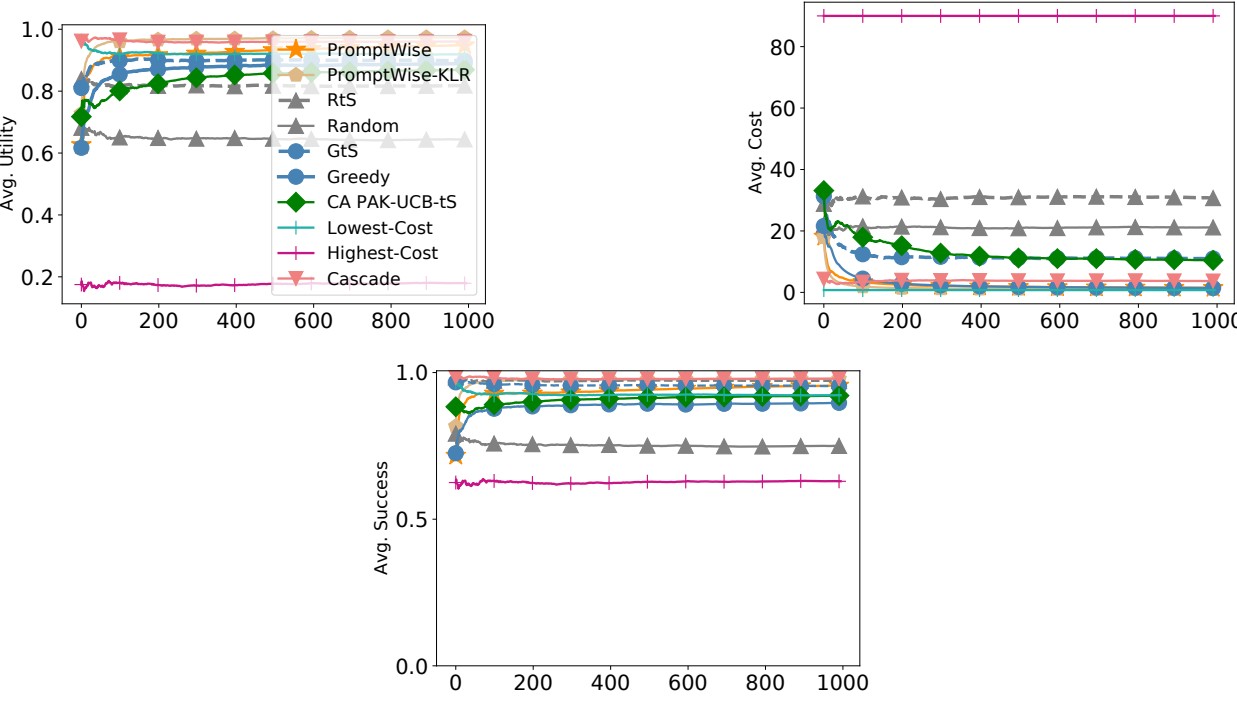

Figure 21: Code completion on the HumanEval benchmark (Task 2): GPT-4o-mini, Claude-3.5-Haiku, Gemini-2.5-Flash-preview, Deepseek-Chat, and Qwen-Plus.

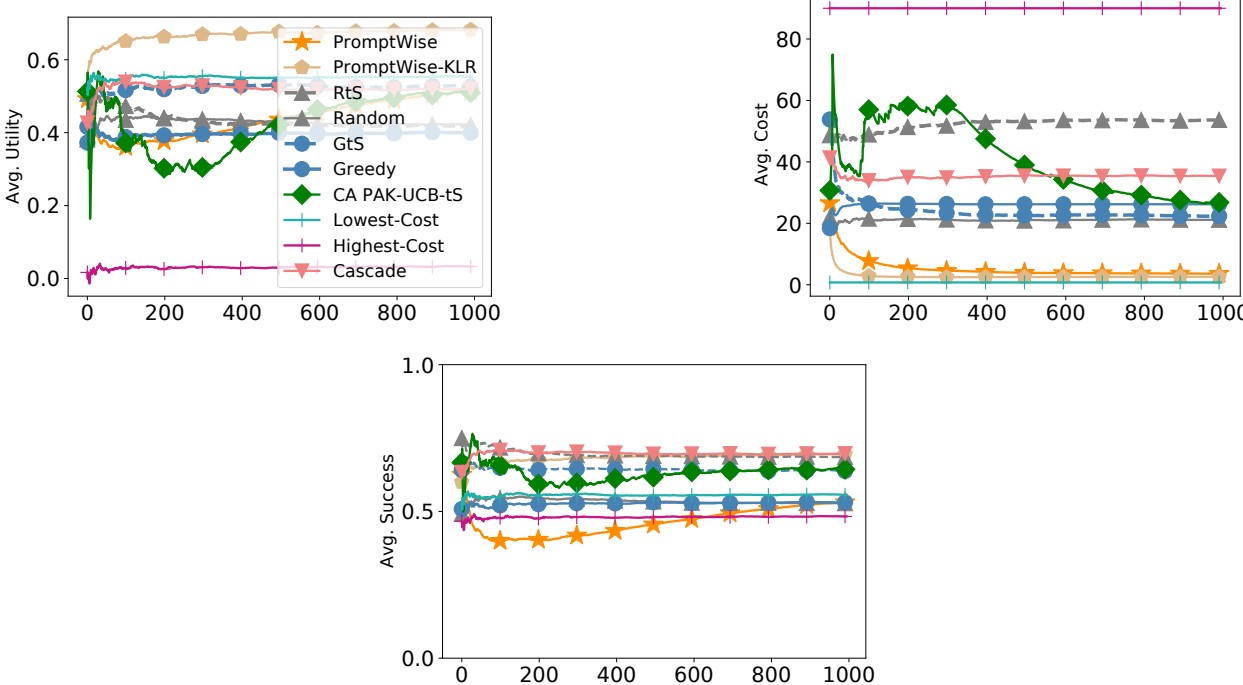

Figure 22: Code completion on BigCodeBench (Task 2): Gemini-2.5-Flash-preview, Deepseek-Chat, Qwen-Plus, GPT-4o, and Claude-Opus-4. Results are averaged over 20 trials.

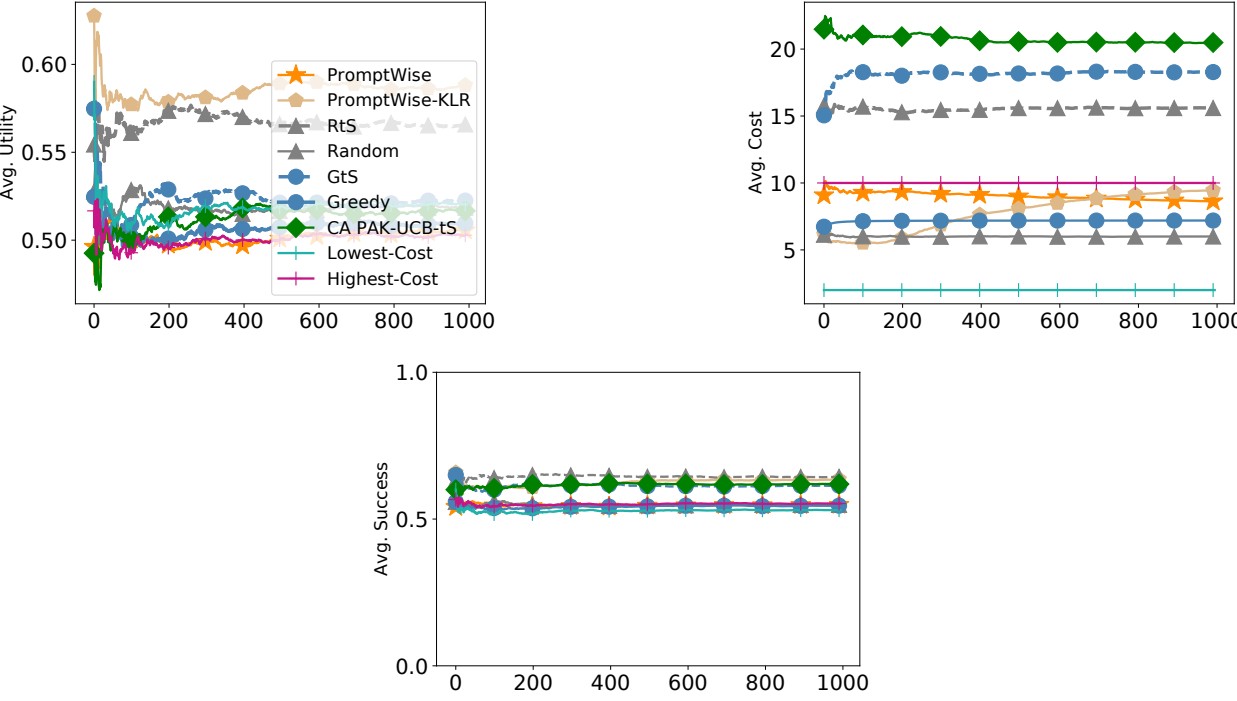

Figure 23: Code completion on BigCodeBench (Task 2): GPT-4o-mini, Claude-3.5-Haiku, Deepseek-Chat, and Claude-Opus-4. Results are averaged over 20 trials.

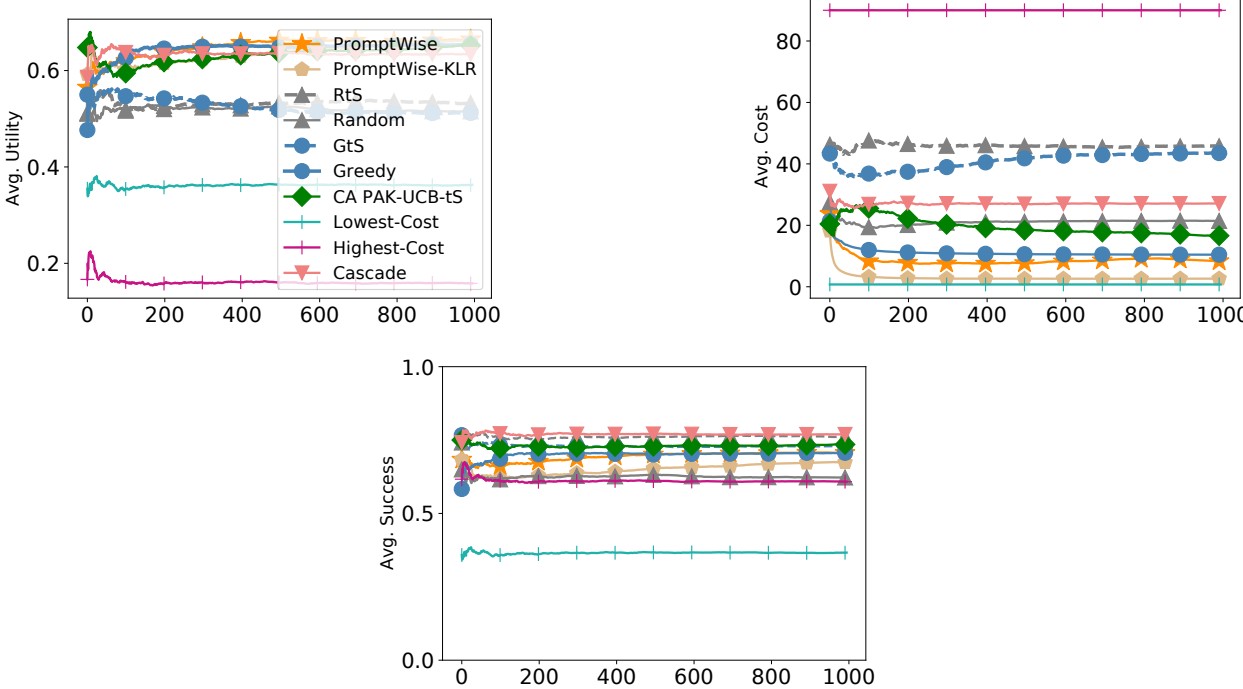

Figure 24: Code translation on the HumanEval-X benchmark (Task 3): Gemini-2.5-Flash-preview, Deepseek-Chat, Qwen-Plus, GPT-4o, and Claude-Opus-4. Results are averaged over 20 trials.

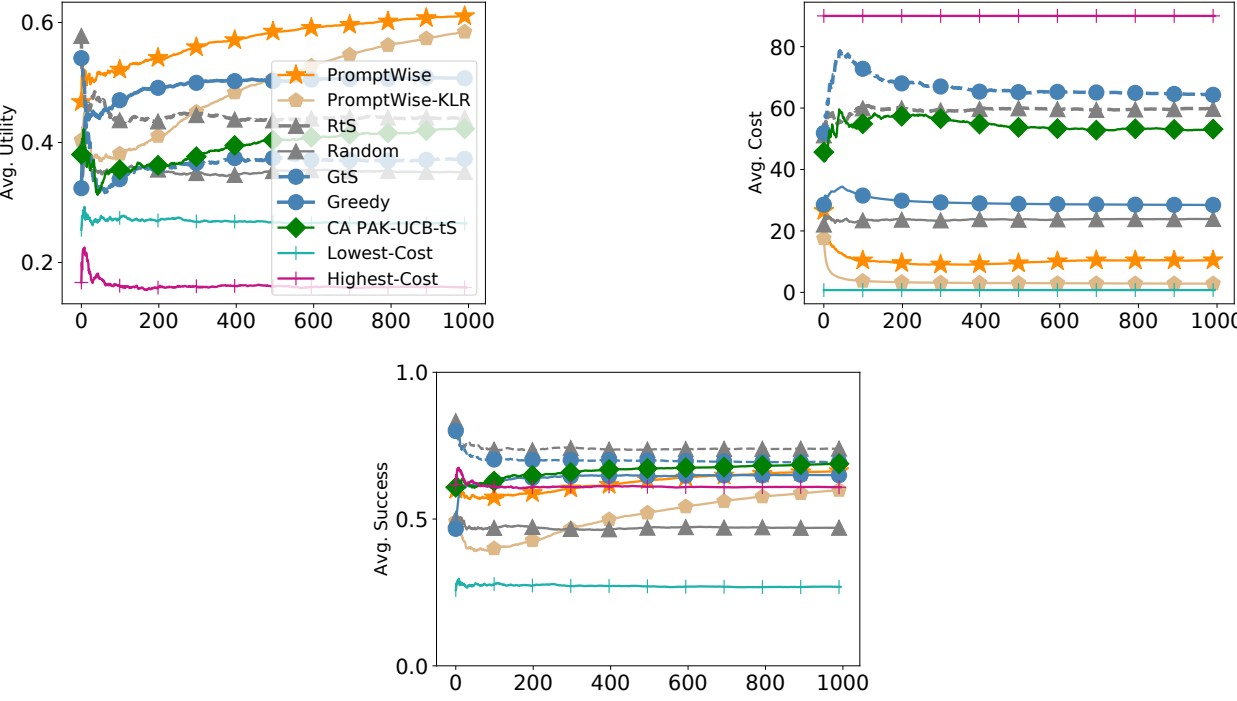

Figure 25: Code translation on the HumanEval-X benchmark (Task 3): GPT-4o-mini, Deepseek-Chat, Claude-3.5-Haiku, and Claude-Opus-4. Results are averaged over 20 trials.

| Prompt Type (MS-COCO) | Expert 1 | Expert 2 | Expert 3 | Expert 4 | Expert 5 |
|---|---|---|---|---|---|
| **Cost** | 0.75 | 1.37 | 1.6 | 12.5 | 90.0 |
| carrot | | | | | |
| dog | | | | | |
| cake | | | | | |
| car | | | | | |
| bowl | | | | | |

Figure 26: Arms in synthetic T2I generation: We set the model costs to align with Tasks 1-3. Prompts are randomly drawn from five categories in the MS-COCO dataset (Lin et al., 2015), including 'carrot', 'dog', 'cake', 'car', and 'bowl'. Each row and column displays the generated images from an arm/expert conditioned to the prompt type. The $i$-th expert is constructed such that it outputs a (clean) image from Stable Diffusion v2 for the first $i$ prompt types, while outputting a white-noise-perturbed image with probability 0.5 for the rest of the prompt types.

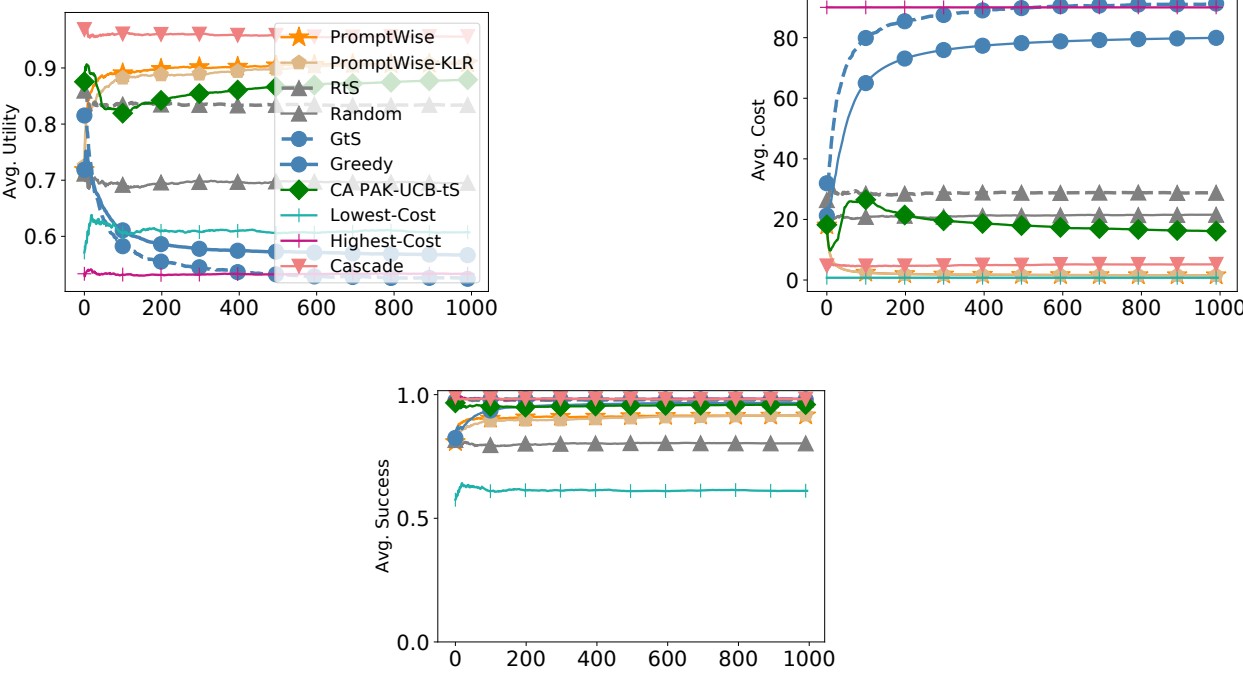

Figure 27: Synthetic Text-to-Image (T2I) Generation (Task 4): The synthetic arms are visualized in Figure 26, where $i$-th arm is an expert in generating images for prompts of the first $i$ types. The reward for an generated image $I$ is defined to be $\mathbb{1}[\text{CLIPScore}(I) \geq 30]$. Results are averaged over 20 trials.

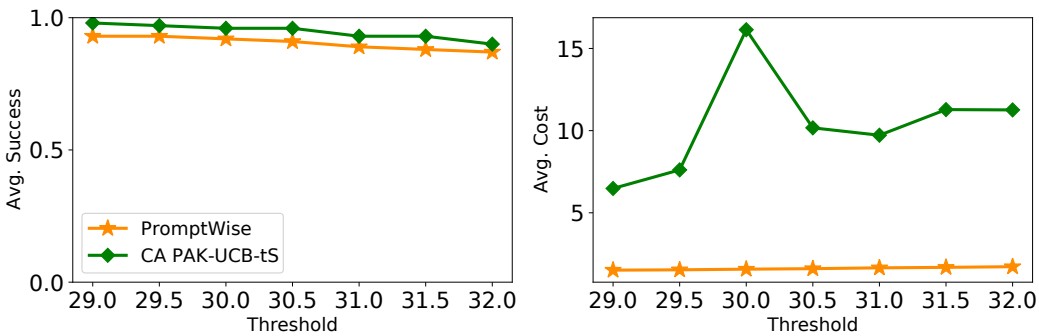

Figure 28: Performance of PromptWise in T2I Generation: Thresholds vary from 29.0 to 32.0.

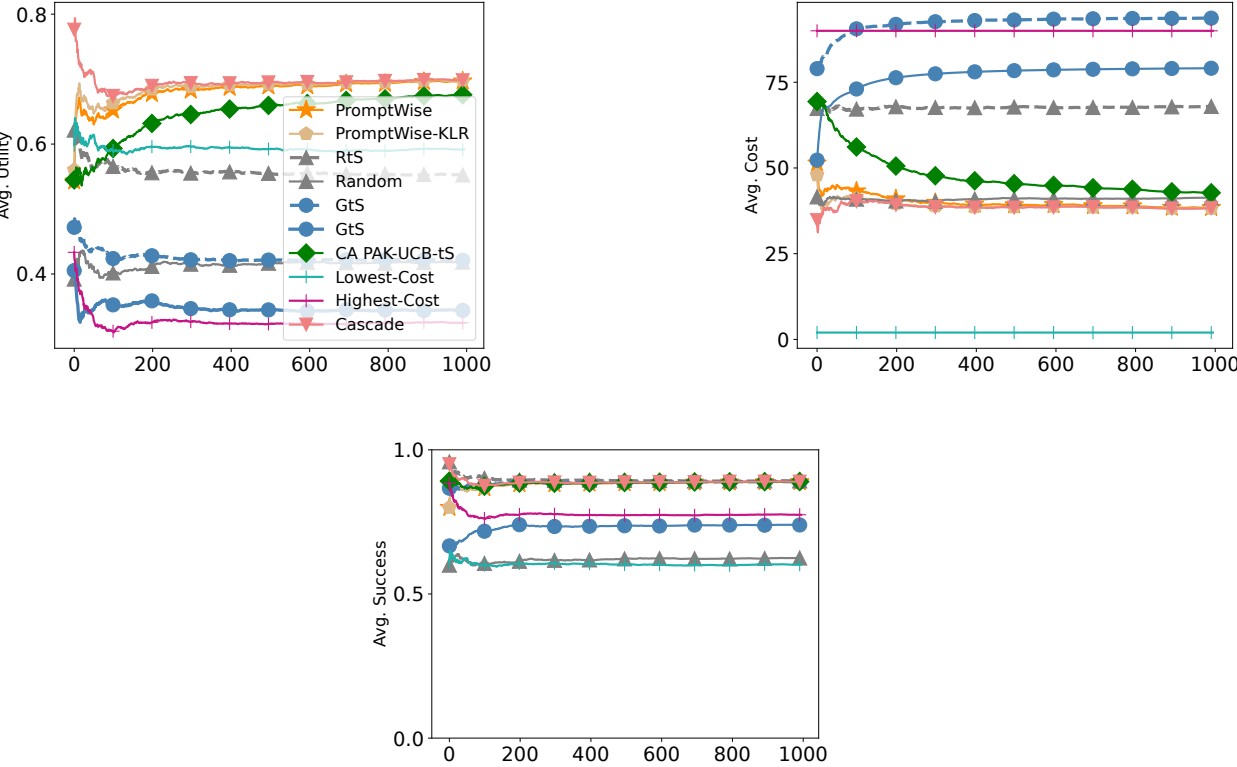

Figure 29: Q&A task on Routerbench: Claude-v2, GPT-4, and Llama-2-70b-Chat.

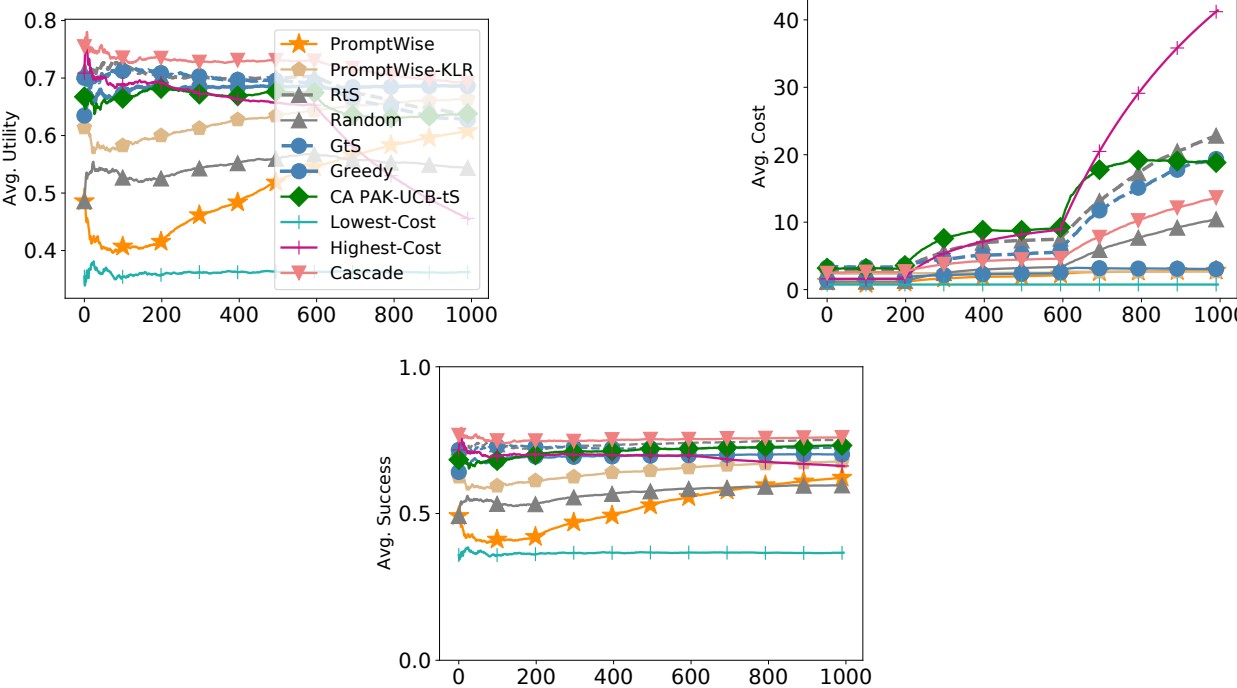

Figure 30: Adaptation to new models on code translation task: Gemini-2.5-Flash-preview, Deepseek-Chat, Qwen-Plus, GPT-4o, and Claude-Opus-4. The model pool contains two LLMs initially, i.e., Qwen-Plus and Gemini-2.5-Flash-preview. In the process, GPT-4o, Deepseek-Chat, and Claude-Opus-4 are added to the pool after each 200 steps. Results are averaged over 20 trials.

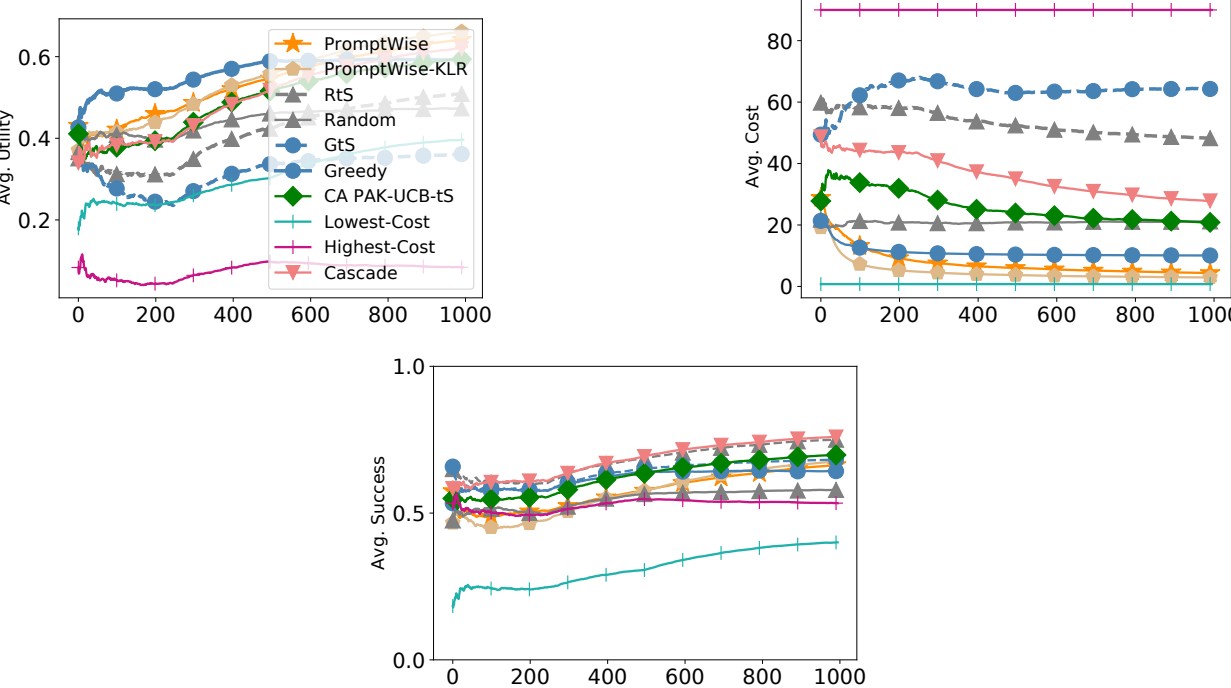

Figure 31: Adaptation to New Prompts: Gemini-2.5-Flash-preview, Deepseek-Chat, Qwen-Plus, GPT-4o, and Claude-Opus-4. Initially, the LLM is asked to solve C++-to-Java translation. Then, Java-to-C++ and Python-to-C++ tasks are introduced after each 250 steps. Results are averaged over 20 trials.

---

**Algorithm 3** PromptWise-KLR

---

**Require:** step $T$, cost parameter $\lambda > 0$, regression dataset $\mathcal{D}_a \leftarrow \varnothing$ for $a \in \mathcal{A}$, $V_a \leftarrow \mathbf{0}_{d \times d}$, round budget $\tau_{\max}$, tuning parameter $\tau_{\exp}$ and $\alpha$, regularization parameter $\beta$, bonus $\mathcal{B}$

1: **for** $a \in \mathcal{A}$ **do**
2:     Collect $\mathcal{D}_a \leftarrow \mathcal{D}_a \cup \{(x_s, r_s)\}_{s=1}^{\tau_{\exp}}$ by pulling arm $a$ (and moving on to the next context) upon observing contexts $\{x_s\}_{s=1}^{\tau_{\exp}}$ drawn from the environment.
3:     Calculate the weight $\widehat{w}_a$ by minimizing the regularized negative

$$\widehat{w}_a \leftarrow \underset{w \in \mathbb{R}^{|\mathcal{D}_a|}}{\arg\min} \left\{ -\sum_{(X,R) \in \mathcal{D}_a} (R \cdot \log \widehat{q}_a(X, w | \mathcal{D}_a) + (1-R) \cdot \log(1 - \widehat{q}_a(X, w | \mathcal{D}_a))) + \beta \|w\|_{\mathbf{K}_{\mathcal{D}_a}}^2 \right\}. \quad (10)$$

4: **end for**
5: **for** $t = |\mathcal{A}|\tau_{\exp} + 1, \cdots, T$ **do**
6:     Observe context $x_t$ and compute the estimated success probability for each arm $a \in \mathcal{A}$:

$$\widehat{q}_a \leftarrow \mu \left( \sum_{(X,R) \in \mathcal{D}_a} \widehat{w}_a^X k(X, x_t) + \alpha \cdot \mathcal{B}(x_t | \mathcal{D}_a) \right). \quad (11)$$

7:     **while** not playing null action $a_0$ **do**
8:         **if** $\max_{a \in \mathcal{A}}\{\widehat{q}_a - \lambda \cdot c_a\} \leq 0$ **or** hitting the round budget **or** receiving a reward of 1 **then**
9:             Play null action $\widehat{a} \leftarrow a_0$ and move on to the next step.
10:         **else**
11:             Play arm $\widehat{a} \leftarrow \arg\min \frac{c_a}{q_a}$ and receive reward $r$.
12:             Update the regression dataset $\mathcal{D}_{\widehat{a}} \leftarrow \mathcal{D}_{\widehat{a}} \cup \{(x_t, r)\}$.
13:             Recompute the MLE $\widehat{w}_a$ and the estimate $\widehat{q}_{\widehat{a}}$ by Equations (9) and (5), respectively.
14:         **end if**
15:     **end while**
16: **end for**

---

