# OpenReview forum: "PromptWise: Extending Contextual Bandits to Multi-Round Prompt Routing for Cost-Efficient LLM Inference"
_TMLR — Under review for TMLR_

### Review · Reviewer_6vjE · 2026-07-18

**Summary Of Contributions:**

The authors introduce PromptWise, an online routing framework for selecting among (language or generative) models while balancing answer quality against cumulative inference cost. The main departure from conventional LLM routers is that it can query multiple models for the same prompt: it may try a cheap model first, only after try the most costly model, and stop once it succeeds or further attempts are not worth the cost.

The authors provide both theoretical evidence and quite large empirical evidence on the ability of PromptWise to work effectively.

The main concern I have is the lack of polishing in some parts of the work, which makes it difficult to better assess the correctness of some claims. Then, I think some simple baselines can be included in the main experiments, to better assess the effectiveness of the approach.
Despite these two shortcomings, I did not mind the work and I think with proper changes the work could be accepted.

**Audience:**

Yes

**Audience Explanation:**

I would say the paper is quite well executed with interesting findings, even though it is not properly polished. Overall, I did not mind reading this work, as it is quite accessible and well framed.

**Broader Impact Concerns:**

It seems like a broader impact statement is currently missing. I think the authors could add one.

**Claims And Evidence:**

Yes

**Claims Explanation:**

The paper seems quite convincing in its arguments, and (as far as I could check) both theoretical evidence and empirical evidence provided by the authors seem sound. However, several parts are not sufficiently clear, in my opinion, and need to be addressed to better evaluate the paper. Please check the requested changes section to see more detailed comments.

**Requested Changes:**

These are my requested changes:

- Please be more precise in the theoretical statements and related proofs:
    -  I am quite confused by how Theorem 1 and Theorem 2 are presented right now. I think the authors should be more precise about what the actual theorem is, as it is not clear what is shown and what is not shown directly. Please clarify better what proves what, eventually adding Theorem 2 directly in the main paper.
    - in Appendix B.2, add a couple of passages to the equation to make it easier for the reader following the proof. Add also a final phrase regarding concavity in $\theta$ (as that one is a first order condition)
    - If I am not mistaken, I cannot see Proposition 2 proof
- A few additional baselines could be considered (they would improve the empirical evidence) or at least explicitly discuss why they are not included:
    - an Oracle baseline (from Prop 1)
    - FrugalGPT [1], which seems to address a very similar problem

-  several additional results seem to be missing (at least no discussion is provided). See Appendix C.1, C.2 (Cost parameter $\lambda$; Round budget $\tau_{\text{max}}$ paragraphs). I think these are results that would improve the strength of the paper.

- I would suggest that the authors provide some sort of statistical uncertainty around point estimates in all the figures without them (e.g., use seaborn lineplot, which provides confidence intervals by default)

- better explain the green and orange lines in Figure 5 (directly in the caption)

- I think the authors could add some references to quite close literature in spirit, such as the one on learning to defer and dynamic model selection. See e.g., [2] for a recent overview of these methods, and [3] for a recent Learning to Defer online variant.

 Finally, I think having so many references to Figures in the Appendix does not improve the paper's flow. Thus, I would suggest that the authors either add some figures in the main text or directly defer the discussion to the Appendix.



[1] - Chen, L., Zaharia, M., & Zou, J. FrugalGPT: How to Use Large Language Models While Reducing Cost and Improving Performance. Transactions on Machine Learning Research (2024)

[2] - Ruggieri, S., & Pugnana, A.  Things machine learning models know that they don’t know. In AAAI (2025)

[3] - Duy, D. H., Montreuil, Y., Meyer, M., Carlier, A., Ng, L. X., & Ooi, W. T. (2026). Online Learning-to-Defer with Varying Experts. In AISTATS (2026)

---

### Review · Reviewer_wpC3 · 2026-07-18

**Summary Of Contributions:**

**The work's contributions are:**
- It identifies a gap in existing cost-aware online scheduling.
- Existing works mostly optimize linear functions of cost and reward, whereas this work considers a linear function of the cost and the maximum reward over rounds.
- The setting of a Cost-Aware Contextual Multi-Armed Bandit is formally defined and analyzed.
- As a solution to the above setting, the PromptWise online algorithm is defined (and two variants are presented). Again, a formal analysis is provided.
- PromptWise and PromptWise-KLR are thoroughly evaluated on textual Sudokus, code completion, code translation, and text-to-image generation.
- The discussion (Sec. 6) extends the analysis to hard tasks, latency-aware optimization, and a dedicated LLM router benchmark.

**Strengths:**
- The work is well written.
- The topic is relevant.
- The formalism is convincing and presented well.
- The experiments are extensive and mostly suitable for supporting the claims.

**Weaknesses:**
- The simplification that costs are independent of the context and that reward feedback can be obtained, both substantially limit the method's applicability.
- The empirical support is difficult to assses to to the lack of aggregation.
- No significance tests on the empirical findings are done.

**Additional Comments:**

**Suggestions:**
- The related work is already rather extensive. However, the task of cost-aware scheduling is much more general. One could consider adopting approaches that, for instance, select simulation fidelity levels in finite element or fluid simulations. This is not a strictly requested change, as I cannot confirm such literature exists yet. If it does not, the similarity could serve as motivation in the introduction and yield an interesting direction for future work.

**Minor Comments:**
- Abstract: The second-to-last sentence seems mostly redundant.
- Sec. 4.1: Clicking on "Algorithm 1" links to "Protocol 1" instead of "Algorithm 1".
- Sec. 4 formatting: The subsection levels are a bit odd (don't reflect the content). My suggestion would be to have three: 4.1 Overview of PromptWise, 4.2 EP-PromptWise, and 4.3 Extension to kernel methods.
- Sec. 4.1.1: The cascade of the three "where ..." suffixes for explaining the formulas is hard to read linearly. Consider breaking it up or (better) reversing the order ot build it up gradually.
- Sec. 5: If one task is text-to-image generation, the text should probably speak of image generation models in addition to LLMs.
- Sec 6, § PromptWise in hard-task scenario: Should reference Tab. 2.
- App. B.2: The last simplification step needs more details to be clear.
- App C.3: The "§ Model costs" is empty. It should reference Tab. 3 and provide the total money spent in the experiments.
- App C.3, Tab 3: The caption should state the date on which the prices were compared. Furthermore, the footnotes in the first column seem to lead nowhere.

**Audience:**

Yes

**Audience Explanation:**

Efficient LLM routing is an important task due to its high monetary relevance. Furthermore, the formal treatment might be applicable to other scheduling tasks, where similar reinforcement learning approaches could be employed.

**Broader Impact Concerns:**

None. The work is good as-is.

**Claims And Evidence:**

Yes

**Claims Explanation:**

**Mostly, however:** (See Requested changes.)
- The claimed superiority of the approach is hard to verify, given the diverse set of results.
- The "Cascade" baseline is rather competitive. Better aggregate results are needed to reliably judge the performance gap.

**Minor Issues:**
- Sec. 3, Remark 1: In general, there are many more quality metrics to coding tasks than passing a test suite. For instance, engineers are typically concerned with aspects such as efficiency, reliability, maintainability, and portability, among other factors.
- Sec. 3, Prop. 1: In practice, the cost of pulling an arm (running an LLM) also depends on the prompt $x$. This should be more clearly marked as a limitation (and avenue for future improvements). How would PromptWise need to change to incorporate this?

The formalisms were correct to the degree I verified. I did not systematically check the Appendix, only parts.

**Requested Changes:**

1. Include the additional baselines mentioned in Sec. 5.1 (§ Baselines) in the main body for easier comparison.
2. Aggregate findings: The entire work evaluates a substantial number of settings (baselines and datasets). Please include an aggregate presentation of the findings, e.g., using win counts. Furthermore, critical difference (CD) diagrams or similar methods can also visualize the statistical tests you should perform on these aggregate findings. This will also show the difference to "Cascade", which is often rather competitive to the presented approaches.
3. Routerbench measured exactly what is required. It should, therefore, be more at the center stage of the numerical evaluation instead of part of the discussion section.

---

### Review · Reviewer_CAxJ · 2026-07-18

**Summary Of Contributions:**

The paper introduces a cost-aware strategy to select the sequence of LLMs for given prompts using a linear or kernel prompt answering probability model. The maximized utility function is the mathematical expectation of binary success minus the weighted sum of costs therefore allowing to learn multi-round strategy differing from single-shot approaches. The proposed strategy can be learned online and is competitive to several empirical alternatives in several types of tasks such as random, greedy or cascade baselines. From the theoretical point of view simplified epoch-updated variant is shown to achieve sublinear regret under independent contexts and some natural assumptions. However the theoretical part is related mainly to infinite-budget oracle and the additional assumptions of the independence of success by consecutive application of the same LLM on the same prompt seems doubtful, the proposed optimistic UCB probability approximation seems mathematically degenerate under low number of training samples and the obtained empirical results doesn’t look really strong compared even to a simple cascade baseline when the query is rooted from cheaper to more expensive LLM until it is not answered satisfactorily.

**Additional Comments:**

Table 1 should contain also utility. The paper probably needs a restructuring in order to avoid the reader switching many times between the main text and the appendices. Also the part of introduction about the Online learning methods for training generative models looks irrelevant to the paper topic.

**Audience:**

Yes

**Audience Explanation:**

Cost-aware routing with sequential attempts is relevant both to researchers and practitioners working with heterogeneous set of LLMs. The proposed in the paper utility function is relevant and the proposed approach deserves formal study. A corrected formulation and adequately documented evaluation could make the work valuable to the contextual-bandit and model-routing communities.

**Broader Impact Concerns:**

probably the proposed algorithm will introduce disparity between the users who propose the prompts of different difficulty

**Claims And Evidence:**

No

**Claims Explanation:**

The superiority of proposed techniques is shown on a limited set of examples and it is not clear how it will be working in practical conditions considering the close or sometimes better results of a cascade LLM selection strategy. For example, in Fig. 3 in Code completion task cascade baseline outperforms simple PromptWise by obtained utility and differs by its kernel version by a statistically small margin. The same situation holds for HumanEval task acc. to Table 1. Also, with 768- or 1536-dimensional embeddings, the reported choice of $\tau_{exp} = 1$ (one initial sample per arm) makes both the learned logistic regression for success probability degenerate and may induce mathematical problems with the exploration bonus term so the proposed algorithm is probably unviable at the conditions reported in the paper. Also I have concerns about the algorithm reproducibility. Also, Appendix C.1 that is referenced in the main text and should contain the comparison between usual and Epoch algorithms is empty.

**Requested Changes:**

Apart from the minuses mentioned above there is a serious flaw and limitation of the proposed approach and its initial assumptions. The problem is that the consecutive runs of the same LLM on the same prompt are not independent in terms of obtained probability (e.g. if GPT 5.4 didn’t answer some prompt correctly and estimated probability is 0.5 it is very possible that this exact prompt won’t be answered during the consecutive runs). Probably that’s the reason why the cascade baseline shows very strong results as it doesn’t use the same model twice. Probably if the authors would address this question both in limitations and in some improved version of the algorithm, it may strongly improve the experimental results. Analogically it looks strange when infinite-horizon oracle will always pull the same unsuccessful arm (until probably the probability model won’t be updated).

The experimental setup should be better documented. The best solution will be to release code, configuration files, prompt sequences, validators, and either the API outputs or a reproducible data-collection procedure.

Address the concerns of too small initialization considering the large embedding size by adding some additional regularization or smart initialization of the weights and matrices in the linear model.

Add stronger baselines from cited literature either with offline learned oracles or finite budget policies.

Appendix C.1 must be filled.